# A New Large †Pachycormiform (Teleosteomorpha: †Pachycormiformes) from the Lower Jurassic of Germany, with Affinities to the Suspension-Feeding Clade, and Comments on the Gastrointestinal Anatomy of Pachycormid Fishes

**Samuel L. A. Cooper** [1,2,*], **Sam Giles** [3,4] , **Holly Young** [5] **and Erin E. Maxwell** [1]

1   Museum am Löwentor, Staatliches Museum für Naturkunde Stuttgart, 70191 Stuttgart, B-W, Germany
2   Department of Paleontology, Hohenheim University, 70599 Stuttgart, B-W, Germany
3   School of Geography Earth and Environmental Sciences, University of Birmingham, Birmingham B15 2TT, UK
4   Department of Earth Sciences, Natural History Museum, Cromwell Road, London SW7 5BD, UK
5   Independent Researcher, Birmingham B15 2TT, UK
*   Correspondence: samuel.cooper@smns-bw.de

**Abstract:** Pachycormiformes is a diverse clade of stem-teleost actinopterygian fishes with a stratigraphic range from the Lower Jurassic (Toarcian) to Upper Cretaceous (Maastrichtian). The Toarcian Posidonienschiefer Formation in SW Germany records the earliest occurrence of †Pachycormiformes in the fossil record, offering unique and crucial insight into the clade's origins and early adaptive radiation in the Early Jurassic. However, Early Jurassic taxa remain poorly studied with the taxonomic diversity and stratigraphic/geographic distributions insufficiently defined, thus masking the early part of this evolutionary radiation. Here, we report a new genus and species of pachycormid fish from the Posidonienschiefer Formation identified by phylogenetic analysis as falling in an intermediate position between *Saurostomus* and *Ohmdenia* at the base of the suspension-feeding clade. The new taxon shows a unique suite of cranial and postcranial characters. Several synapomorphies of the suspension-feeding clade, notably, the morphology of the hyomandibula, elongation of the skull, and reduced squamation are shared with the new taxon. The intestinal tract is exceptionally preserved, providing one of the most complete examples of pachycormid gastric anatomy. A comparison of the gastrointestinal anatomy of the new genus with other pachycormiforms indicates extensive taxonomic variation within the clade, in the configuration of both the midgut and spiral valve, potentially related to trophic divergence. The results highlight an underestimated high diversity and the rapid acquisition of trophic specializations in Pachycormiformes much earlier in the clade's evolution than previously considered.

**Keywords:** Pachycormiformes; Toarcian; Posidonienschiefer; paleoecology; suspension-feeding; exceptional preservation; spiral valve; Jurassic; Actinopterygii

## 1. Introduction

### 1.1. Pachycormiform Evolution

†Pachycormiformes is a monophyletic clade of Mesozoic actinopterygian fishes that holds an important position in the Holostei to Teleostei transition. The only included family, †Pachycormidae, is known exclusively from marine strata ranging from the Early Jurassic (Early Toarcian) discontinuously to the Maastrichtian of the Late Cretaceous [1–6]. Globally speaking, their distribution and paleoecology are poorly documented, and little is presently understood regarding the macroevolutionary trends influencing their diversity throughout the Mesozoic. Unfortunately, a large portion of pachycormiform material in museum collections is poorly identified, with material quite often being lumped into historic genera (e.g., †*Pachycormus* and †*Hypsocormus*). Early works lumped material from

numerous localities and ages into these poorly diagnosed genera, broadening taxonomic diagnoses and stratigraphic ranges [7–10]. Many museum curators followed this flawed taxonomy, especially for Early Jurassic (Toarcian) materials based on taxa said to be present in Hauff and Hauff's [11] *Das Holzmadenbuch* [12–15]. A recent renaissance of interest in pachycormid research over the last few decades has made progressive steps toward resolving some of these issues, with many historic genera now better defined and several new taxa formally described [5,6,13–17]. Nonetheless, the majority of these studies have focused on stratigraphically younger pachycormids, ranging from the Middle Jurassic to Late Cretaceous [5,6,13–15,17–21], with little research attention directed towards Early Jurassic taxa [1,22–24]. Early Jurassic pachycormid research has unfortunately been heavily biased towards the type taxon *Pachycormus macropterus* (de Blainville, 1818 [25]) [1,22,23,26–28], with very few other Early Jurassic taxa receiving detailed study (see Reference [16] for †*Ohmdenia* and Reference [24] for †*Saurostomus*). There remain numerous outstanding questions in pachycormid evolution, particularly those surrounding the origins and early radiation of the group during the Early Jurassic.

The Lower Toarcian (Lower Jurassic) Posidonienschiefer Formation of southern Germany (Figure 1)—a black shale Konservat Lagerstätte—records the oldest stratigraphic occurrence of Pachycormiformes in the fossil record, with the single oldest specimen, a mostly complete skeleton of the transitional suspension-feeder †*Saurostomus esocinus* marking the family's emergence in the *paltum* Subzone at the base of the Toarcian [24]. From their first appearance in this formation, pachycormids were already taxonomically diverse and morphologically disparate, with at least six genera represented: *Pachycormus* [25]; *Saurostomus* Agassiz, 1843 [29]; *Euthynotus* Wager, 1860 [30]; *Sauropsis* Agassiz, 1843 [29]; *Ohmdenia* Hauff, 1953 [31]; and *Haasichthys* Delsate, 1999 [32]. The seemingly high diversity of pachycormids at their first appearance in the fossil record, and derived traits of *Saurostomus esocinus* (e.g., loss of infraorbitals and suborbitals) compared to co-occurring basal taxa (e.g., *Euthynotus* spp. [33]), strongly suggests that the group's origin predates the Toarcian.

Here, we describe a new genus and species of Early Jurassic pachycormiform from the Posidonienschiefer Formation of Holzmaden (Baden-Württemberg, Germany), bringing the total number of Early Jurassic genera to seven, a more diverse pachycormiform fauna even than that of the Late Jurassic Plattenkalks [15]. We discuss the phylogeny and paleoecology of the new taxon in the context of the clade's early adaptive radiation. The soft tissue anatomy of the gastrointestinal tract is also discussed with particular emphasis on the spiral valve organ in pachycormid fishes.

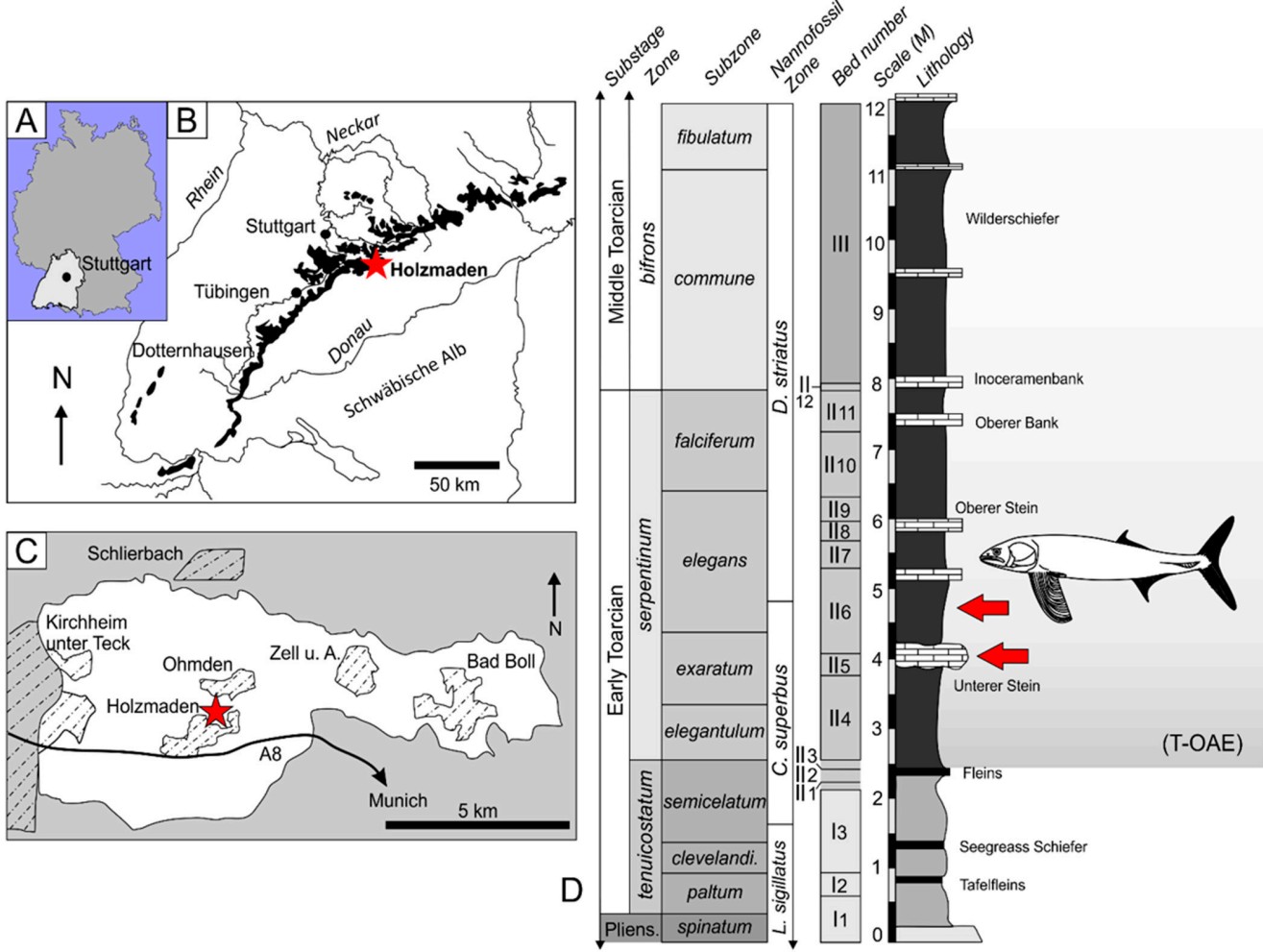

**Figure 1.** Locality maps and stratigraphic column of the Posidonienschiefer Formation: (**A**) Map of Germany with the state of Baden-Württemberg highlighted. (**B**) Stylized map of Baden-Württemberg, showing regional outcrops of the Posidonienschiefer Formation (black) in the region. The study locality of Holzmaden is indicated with a star. (**C**) Simplified map of the Holzmaden region, with the studied locality, within the town of Holzmaden, indicated with a star (modified from Maxwell et al. [34]). (**D**) Stratigraphic log of the Posidonienschiefer Formation at Holzmaden, with the two occurrences of the new pachycormid in beds II5 (*exaratum* subzone) and II6 (*elegans* subzone) indicated by the red arrow icons. Log redrawn and modified from Maxwell and Vincent [35].

*1.2. Geological Context*

The Toarcian (Lower Jurassic) Posidonienschiefer Formation (syn. Posidonia Shale) is a marine black shale Konservat Lagerstätte, world renowned for its exceptionally preserved vertebrate and invertebrate fossils. The black shale deposition was laterally extensive in the Toarcian, covering southwestern and northwestern Germany, northern Switzerland, northwestern Austria, southern Luxembourg (regionally called the Schistes Bitumineux), France, and the Netherlands [36–42]. Of special importance here are the outcrops in Baden-Württemberg (southwestern Germany), particularly at the heavily sampled sites of Holzmaden, Ohmden, Bad Boll, and Zell unter Aichelberg (Figure 1B,C), where numerous scientifically valuable specimens have been collected from for almost two centuries (e.g., [43]) and at the more recently quarried locality of Dotternhausen-Dormettingen [44]. Formally named by Quenstedt [45], the formation name derives from the ubiquitous fossil oyster '*Posidonia bronni*' (synonym of *Bositra buchii* and *Steinmannia bronni*; see Reference [42]), which characterizes a key portion of the mollusk faunal component in the formation. The Posidonienschiefer Formation is well constrained to the Lower Toarcian

based on ammonite zonal fossils, with the section divided into three zones (*Dactylioceras tenuicostatum*, *Harpoceras serpentinum*, and *Hildoceras bifrons*) with several subzones [46] (Figure 1D). The formation comprises finely laminated oil shale mudstones intercalated with bituminous limestones and marls with localized nodular horizons [37,38,47]. Black shales are the dominant lithology in the formation, composed predominantly of detrital clay minerals, organic matter (OM), and pyrite, with beds displaying various ratios of thickness, with exposures around Holzmaden and Dotternhausen measuring between 5 and 14 m thick [38,46,48,49]. Exceptional fossil preservation is attributed to a combination of periodical sea-floor anoxia hindering decay, and soupy substrates promoting rapid burial and early diagenesis [37,43,50,51]. The Posidonienschiefer is divided into three parts (εI–εIII), with each part comprising beds designated by Arabic numerals following the historical numbering scheme implemented by Hauff [10]. Vertebrate fossils are discontinuously distributed across the different beds in the Holzmaden region [10,35], with pachycormid fishes in particular being well distributed, although not continuously across almost all beds [10,11,24].

*Institutional abbreviations*. **BRLSI** = Bath Royal Literary and Scientific Institute, Bath, UK; **BSPG** = Bayerischen Staatssammlung für Paläontologie und Geologie, Munich, Germany; **GPIT** = Paläontologische Sammlung der Universität Tübingen, Tübingen, Germany; **JME** = Jura-Museum Eichstätt, Germany; **KUVP** = University of Kansas Museum of Natural History, Lawrence, Kansas, USA; **NHMUK** = Natural History Museum, London, London, UK; **SMNS** = Staatliches Museum für Naturkunde Stuttgart, Stuttgart, Germany.

*Nomenclatural acts*. This published work and the nomenclatural acts it contains have been registered in ZooBank: urn:lsid:zoobank.org:act:B2B56C1D-D753-4731-9C86-26DBD492487C.

## 2. Materials and Methods

### 2.1. Examined Material

During recent examinations of the Posidonienschiefer pachycormids in the SMNS, we identified two specimens representing a new taxon with characteristics distinct from all other described taxa from the formation (see diagnosis). The first specimen, SMNS 15815—designated here as the holotype—comprises a near complete skeleton from the *exaratum-elegans* Subzone (Early Toarcian) of an unspecified quarry within the classic locality of 'Holzmaden'. SMNS 15815 was originally collected and prepared from the underside by Bernard Hauff sometime either during or prior to the early 1920s. The specimen was then purchased directly from Hauff's workshop by Richard Heilner of Stuttgart, Generaldirektor of Deutschen Linoleumwerke AG, who donated the specimen to the former Württemberg Natural History Collection (now the Staatliches Museum für Naturkunde Stuttgart) in the Autumn of 1927 [52]. The slab containing the specimen is 122 × 54 cm and 6 cm deep and remains mounted within its original wooden frame—a relic from its original exhibition display prior to WWII in the former museum gallery on Neckarstrasse (Württemberg Natural History Collection). An undated photograph in the archives of the SMNS shows the specimen displayed in the old museum gallery prior to the building's destruction during WWII (Figure S1). Given the excellent condition of the specimen, albeit some minor damage to the frame and original specimen label, the fossil was evidently evacuated from the old museum gallery for safeguarding sometime prior to the building's destruction in 1944, which saw almost all of its remaining exhibits destroyed in a fire [53,54].

A second specimen comprising a partial skull and pectoral fin (SMNS 56344) is consistent with this new taxon and was donated to the SMNS in 1947 by one T. Hermann. The Hermann specimen originates from the *exaratum* Subzone of 'Holzmaden' and comprises a split nodule, with the counterpart unaccounted for. Following a thorough search in other institutions with good Lower Jurassic fish collections (e.g., NHMUK; GPIT; Urweltmuseum Hauff) we failed to identify any further material of this new taxon and, thus, all anatomical and stratigraphic data are based solely on the two SMNS specimens.

Specimens were examined first hand using a Wild Heerbrugg ×50 microscope for detailed observations. On account of the holotype's large size, it could not fit under a microscope and, therefore, we examined this specimen using macrophotography and a 10X magnification hand lens. Photographs were taken using a Nikon Series DMC-FZ72 camera with a compact 60X optical macro-lens, with a vertical tripod used to ensure all photographs were taken exactly at 90° vertical from the bedding surface. For accurate measurements we used a 3 m tape measure and a digital caliper, with the values rounded to the nearest 0.5 mm.

For the comparative anatomy, we examined numerous materials from a range of different Jurassic pachycormid taxa, including all *Saurostomus esocinus* specimens listed in Cooper and Maxwell [24]: table 1). The additional taxa examined included:

Early Jurassic taxa:
†*Pachycormus macropterus* (de Blainville, 1818) [25]—SMNS 1818; SMNS 4204; SMNS 89547; SMNS 56230; BRLSI.1297; BSPG 1940-I-6;

†*Pachycormus* sp.—BRLSI.1838; BRLSI.1834;

†*Ohmdenia multidentata* Hauff, 1953 [31]—GPIT-PV-31531 (Holotype);

†*Sauropsis veruinalis* White, 1925 [55]—NHMUK PV P.13007 (Holotype); NHMUK PV P.13006 (paratype); SMNS 53988; SMNS 96618;

†*Euthynotus* spp.—SMNS 89547; SMNS 87432; SMNS 54785;

†*Saurostomus esocinus* Agassiz, 1843 [29]—see Reference [24]: table 1) for a full list of the examined material.

Middle Jurassic taxa:
†*Sauropsis*? sp. (Oxford Clay)—NHMUK PV P. 7568;

†*Hypsocormus leedsi* Woodward, 1889 [56]—NHMUP PV P. 6913 (Holotype);

†*Orthocormus*? *tenuirostris* (Woodward, 1889) [56]– NHMUK PV P. 10579; NHMUK PV P. 10906;

†*Martillichthys renwickae* Liston, 2008 [13]—NHMUK PV P. 61563 (Holotype) (Supplementary File S1: Figure S2);

†*Leedsichthys problematicus* Woodward, 1889 [56]—NHMUK PV P. 1823;

†Pachycormidae indet—NHMUK PV P. 61397;

Late Jurassic taxa:
†*Hypsocormus insignis* Wagner, 1863 [57]—BSPG AS VI 4 (Holotype); SMNS 56650;

†*Hypsocormus posterodorsalis* Maxwell et al., 2020 [15]—GPIT/OS/00836 (Holotype);

†*Orthocormus* sp. (Nusplingen)—GPIT/OS/1302;

†*Simocormus macrolepidotus* Maxwell et al., 2020 [15]—NHMUK PV P. 6011 (Holotype); SMNS 96988/4;

†*Orthocormus roeperi* Arratia and Schultze, 2013 [14]—BSPG 1993 I 22 (cast of Holotype);

†*Sauropsis longimanus* Agassiz, 1843 [29]—BSPG AS VII 1089 (Holotype);

†*Sauropsis* sp.—BSPG 1964 XXIII 525; BSPG 1977 IX 1;

†*Asthenocormus titanius* (Wagner, 1863) [57]—BSPG 1987-I-51; BSP1951 XVI 1.

Cretaceous taxa:
†*Rhinconichthys taylori* Friedman et al., 2010 [5]—NHMUK PV OR 33219 (Holotype);

†*Protosphyraena ferox* Leidy, 1857 [58]—NHMUK PV P. 5630;

† *Protosphyraena*? *Stebbingi* Woodward, 1909 [59]—NHMUK PV P. 11216 (Holotype);

†*Protosphyraena compressirostris* Woodward, 1895 [60]—NHMUK PV P. 5631;

†*Protosphyraena minor* (Agassiz, 1843) [29]—NHMUK PV. 4078;

†*Protosphyraena brevirostris* Woodward, 1895 [60]—NHMUK PV P. 7253 (Holotype); NHMUK PV P. 7252.

### 2.2. Anatomical Terminologies

The nomenclature of cranial bones is based on a proposed homology of the actinopterygian skeleton [61,62]. The nomenclature of specialized structures in †Pachycormiformes (e.g., rostrodermethmoid and cranial boss) follows Lambers [1] and Mainwaring [27], with

the nomenclature for elements of the caudal fin following Arratia and Lambers [63] and Arratia [64]. The pachycormid pectoral fin morphology follows the revised classification by Liston et al. [65], with specialized morphological differentiation in the pectoral fin shape of *Saurostomus esocinus* (morphologies 1 and 2), following the parameters outlined by Cooper and Maxwell [24]. The elements of the vertebral column are defined based on Schultze and Arratia [66,67] and Grande and Bemis [68]. The fulcra patterns and morphologies in the fins follow Arratia [64,69] and Schultze and Arratia [70], with terminologies for scales based on Schultze [71]. The anatomical landmarks defining the measurements of the pachycormid skeleton were updated from Cooper and Maxwell [24] to incorporate additional measurements.

### 2.3. Phylogenetic Analysis

We tested the phylogenetic position of SMNS 15815 within Pachycormiformes by performing two separate phylogenetic analyses using different matrices, with parsimony as the optimization criterion. The first analysis was based on the matrix developed by Friedman [16], of which we retained the original 125 characters; 3 additional characters included were introduced by Cooper and Maxwell [24] and a new character, the presence/absence of a pre-caudal scaly keel (see Supplementary Files S1 and S2), was added, bringing the total to 129 characters. The addition of a new taxon increased the number of taxa to 30. For the second analysis, we used the matrix of Gouiric–Cavalli and Arratia [6], which included one continuous and 185 discrete characters and 48 taxa. All previous characters and character states were retained from the original analysis, as were all taxa with the exception of '*Saurostomus esocinus* Stuttgart', which we removed, as we deemed this taxon to be synonymous with *Saurostomus esocinus* (See Supplementary Files S1 and S3). We updated the scoring of *Saurostomus esocinus* based on recent observations and revised the diagnosis of this taxon [24]. We used the same programs and parameter settings in both of our analyses. Both phylogenetic analyses were conducted using TNT software version 1.5 [72]. The matrix for each was imported from Mesquite and conducted as a New Technology Search with the Ratchet algorithm set to 100 iterations and the minimum length recovered set to 50 times. The random seed was set to 10, and drift was also enabled. The Max trees was increased to 1000 to prevent tree buffer overflow. For the Friedman matrix, the six most parsimonious trees (MPTs) were recovered at a length of 253 and were then combined using a strict consensus (Figure S3). The tree was then resampled using the jackknifing ($p = 36$) function to test clade stability (Figure S4). Lastly, in order to identify problematic or highly unstable taxa, we compared the data using an agreement subtree (standard default settings), with the pruned taxa removed from the cladogram (Figure S5). The same methods were also applied to the Gouiric–Cavalli and Arratia matrix [6], which recovered five parsimonious trees that were also combined into a strict consensus (Figure S6) and clade support was also evaluated using jackknifing (Figure S7) and was resampled as an agreement subtree (Figure S8).

### 2.4. CT Scanning and Visualization

BRLSI.1383, BRLSI.1384, and BRLSI.1297 were CT scanned at the XTM Facility, Palaeobiology Research Group, University of Bristol. BRLSI.1383 and BRLSI M.1384 were each scanned in a single stack, and BRLSI.1297 in two stacks that were later stitched together. All specimens were scanned using a rotating tungsten target and the following settings:

- BRLSI.1383 and BRLSI.1384: 225 kV, 284 uA; 63.9 W; 2.83 s exposure; 3 mm Cu filter; 3141 projections with 4 frames averaged per projection; voxel size 46.43 µm;
- BRLSI.1297: 225 kV, 519 uA, 116.8 W; 1.42 s exposure, 2.5 mm Cu filter; 3141 projections with 4 frames averaged per projection; voxel size 90 µm.

After scanning, the data were segmented in Mimics v.19 (http://biomedical.materialise.com/mimics; Materialise, Leuven, Belgium). Surface meshes were then exported into and imaged in Blender v.2.79 (http://blender.org; Stitching Blender Foundation, Amsterdam, the Netherlands).

## 3. Results

*3.1. Systematic Paleontology*

> Sub-Class ACTINOPTERYGII Cope, 1887 [73];
> Division TELEOSTEOMORPHA Arratia, 2001 [74];
> Order PACHYCORMIFORMES Berg, 1937 [75];
> Family PACHYCORMIDAE Woodward, 1895 [60];
> Subfamily HYPSOCORMINAE Vetter, 1881 [76].

**Definition.** All pachycormids more closely related to *Hypsocormus insignis* than to *Asthenocormus titanius*.

**Remarks.** Phylogenetic analysis reliably resolved two clades within Pachycormidae [5,6]: the so-called 'toothed clade', containing the core genera *Kaykay, Orthocormus, Hypsocormus*, and *Protosphyraena* [17], and the suspension-feeding clade, containing all edentulous pachycormids, including *Asthenocormus, Leedsichthys, Martillichthys, Rhinconichthys*, and *Bonnerichthys*. Despite their core members being mid-Jurassic to Cretaceous in age, the clades diverged in the Early Jurassic, prior to the evolution of suspension-feeding [16,24] and, thus, teeth are present in the basal members of both lineages. In order to facilitate discussion of toothed pachycormids that are not part of the 'toothed clade', we opt to name the two lineages at the subfamily level.

Vetter ([76] p. 90) named two pachycormid subfamilies: Sauropsinae and Hypsocorminae. He defined the latter as including both *Hypsocormus* and *Pachycormus* to the exclusion of *Euthynotus* and *Sauropsis*, and later in the same contribution included *Asthenocormus* within Hypsocorminae. Thus, the original conception of Hypsocorminae was somewhat different than what we are proposing here, and it included the last common ancestor of the toothed and suspension-feeding clades. However, we propose to restrict the usage of this subfamily to those pachycormids that are more closely related to *Hypsocormus* than to the suspension-feeding clade based on the phylogenetic structure of Pachycormidae.

Included genera: *Hypsocormus, Orthocormus, Simocormus, Kaykay, Protosphyraena*, and *Australopachycormus*.

Subfamily ASTHENOCORMINAE nov.

**Definition.** All pachycormids more closely related to *Asthenocormus titanius* than to *Hypsocormus insignis*.

**Remarks.** Asthenocorminae includes the edentulous 'suspension-feeding clade' but also those toothed pachycormids resolved as early-diverging members of the suspension-feeding lineage (*Saurostomus* and *Ohmdenia*). *Pachycormus* is phylogenetically unstable; based on the preferred phylogeny presented here, this genus is the earliest diverging asthenocormine, but this relationship is very poorly supported.

Included genera: *Pachycormus, Saurostomus, Germanostomus* gen. nov., *Ohmdenia, Martillichthys, Leedsichthys, Asthenocormus, Rhinconichthys*, and *Bonnerichthys*.

**GERMANOSTOMUS** gen. nov.
*ZooBank LSID.* urn:lsid:zoobank.org:act:B2B56C1D-D753-4731-9C86-26DBD492487C
*Type species.* **Germanostomus pectopteri** gen. et sp. nov.; see below.
*Diagnosis.* As for the type and only species, *G. pectopteri* (below).
*Etymology.* Genus named for the Southwestern Germanic basin where the taxon originates; with the suffix *stomus* (Greek) for 'mouth' deriving from the closely related genus *Saurostomus*.

**Remarks.** *Germanostomus* is characterized by a rostrodermethmoid contributing to the dentigerous anterior margin of the upper jaw and separating the nasals; a shallow coronoid process; a lower jaw extending behind the orbit; and pectoral lepidotrichia which are un-

segmented and distally bifurcate asymmetrical of the joints. This combination of characters is consistent with the placement of this genus within Pachycormiformes [1,5,16,27]. The two large plate-like suborbitals (secondarily lost in †*Saurostomus esocinus* and more derived asthenocormines [24]) are absent, suggesting a close affinity of *Germanostomus* with the suspension-feeding pachycormids.

*Germanostomus pectopteri* gen. et sp. nov.
*ZooBank LSID*: urn:lsid:zoobank.org:act:B2B56C1D-D753-4731-9C86-26DBD492487C

Figures 2–4, 5A,D, 6, 7A–D,G and 8.
1927.—*Pachycormus* sp., Berckhemer [52], pp. 22–24.
1930.—*Pachycormus* sp., Schwenkel [77], figure 6.
1979.—*Pachycormus* sp., Urlichs et al., [78].
1985.—*Pachycormus* sp., Jäger [79], figure 26.
2019.—*Pachycormus* sp., Liston et al., [65], table 1.
2022.—*Saurostomus* sp., Cooper and Maxwell [24], figure 13d.
*Holotype*. SMNS 15815—Imperfectly complete, moderately well-articulated skeleton prepared from the underside in the left ventrolateral view (εII 6), Figures 2–7 and 8A.
*Paratype*. SMNS 56344 a partial pectoral fin associated with the gut contents, and the disarticulated elements of the cranium is preserved in the nodule part (εII 5), Figure 8B.
*Diagnosis*. Skull elongated and roughly four times longer than deep, contributing approximately 24% of the total body length. Premaxilla elongate, approximately half the length of the maxilla and contributing 33% of the upper jaw length excluding the rostrodermethmoid (≤17% in *Saurostomus esocinus*); premaxilla occupying entire ventral margin of the antorbital (mostly occupied by the maxilla in *Saurostomus esocinus*); premaxilla holds at least eight principal teeth (no more than four in *S. esocinus*). Lower jaw elongate with length to depth ratio of 5:1 (4:1 in *S. esocinus*); dentary elongate, laterally ornamented, overbite absent. Teeth well-spaced, conical with straight crowns and lacking basal grooves. Infraorbitals and suborbitals absent. Hyomandibula tall, rectangular, and gently waisted lacking expanded distal ends, and possessing both a medioanterior and medio-posterior lamina with straightened vertical margins, opercular process absent (hyomandibula is hourglass-shaped with expanded distal bone margins and well developed opercular process in *Saurostomus esocinus*). Cranial bones weakly ornamented. Preopercle tall and splint-like with shallow ventral-posterior fan ornamented with vertical grooves; opercle large, trapezoidal; subopercle large, rectangular. Cleithrum robust and 'L'-shaped (similar to *Ohmdenia* and *Saurostomus*); seven proximal radials, anterior radials shorter with wider distal terminations, posterior radials taller and slimmer; propterygium small, articulated between first and second pectoral lepidotrichia. Pectoral fin an inverted 'D'-shape with anterior three rays fused into a stiff, curved rod forming an obtusely curved leading edge encompassing the entire ventral margin of fin; pectoral fin lacking fulcra, composed of approximately 28 distally bifurcating lepidotrichia which terminate at the same exact plane 90° to the proximal edge forming a perfectly straightened posterior fin edge. Posterior fillet highly reduced, comprising a shallow lobe-like projection of the proximal-posterior fin margin, and composed of the last few lepidotrichia not attached to the ossified radials. Vertebral column very poorly ossified, arcocentra not extending around notochord, chordacentra absent. Anal fin strongly falciform and set close to caudal fin, anal lepidotrichia medially segmented prior to distal bifurcation. Basal dorsal scute rhombic with a prominent medial-dorsal ridge (seen in *Ohmdenia*) and anteriorly ornamented with coarse striations (similar to both *Saurostomus* and *Ohmdenia*) but without an anterior notch (unlike in *Saurostomus* and †*Martillichthys*). Caudal fin homocercal and deeply forked. Hypural plate tall and narrow with a straight anterior margin and a small hypural process set roughly at the midline. Hypaxial basal fulcra unpaired and extending along approximately one-quarter of the ventral caudal leading edge. Fringing fulcra along hypaxial leading edge small, type-A, triangular (large in *Saurostomus*). Pre-caudal scaly keel present, composed of at least ten enlarged scales per side. Squamation reduced, very poorly ossified (more so

than in *Saurostomus*), scales rhombic. Intestine straight with a sigmoidal curvature and a deeper foregut region compared to the hindgut. Hindgut composed of an elongated spiral valve set close to the anus with a minimum of 16 rotations.

*Type locality.* 'Holzmaden', Baden-Württemberg, SW Germany.
*Type horizon.* Posidonienschiefer Formation, Lias Epsilon II 6, *exaratum–elegans* Subzones, *serpentinum* Zone, Early Toarcian, Lower Jurassic.
*Stratigraphic range. exaratum–elegans* Subzones, middle of *serpentinum* Zone, presently restricted to the locality of 'Holzmaden', Baden-Württemberg, Germany.
*Etymology. pectopteri = pecto* (pectoral) with *pteros* (wing), referencing the highly distinctive wing-shaped hydrofoil morphology of the large pectoral fin.

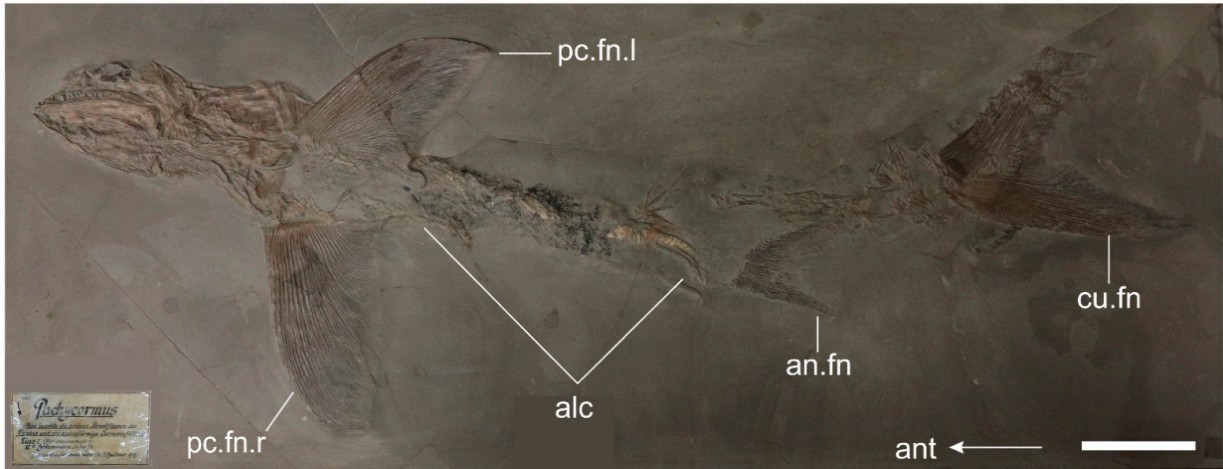

**Figure 2.** *Germanostomus pectopteri*, holotype. SMNS 15815 prepared from the underside revealing the skeleton in left lateroventral view. Note the contrast in articulation quality between the anteroventral and posterodorsal regions of the skeleton, implying a head-first burial within the soft substrate. alc, alimentary canal; an.fn, anal fin; ant, direction towards anterior of fish; cu.fn, caudal fin; pc.fn, pectoral fin (l, left; r, right). Scale bar = 100 mm.

### 3.2. Description

#### 3.2.1. General Features

A large, fusiform pachycormid with a total length (TL) of 1060 mm, standard length (SL) of 850 mm, body length (BL) of 605 mm, head length (HL) of 245 mm, and a mandible length (ML) of 125 mm (Table 1). The head is much longer than deep, and contributes to just under one-quarter of the total body length. Likewise, the mandible is elongated, with a length-to-depth ratio greater than in the largest examples of *Saurostomus esocinus* but only half that of *Ohmdenia multidentata*. The pectoral fin is large and superficially 'D'-shaped, with a posteriorly straight margin. The vertebral column is poorly ossified, with simple neural and haemal arches articulating on the notochord. The caudal fin is homocercal and forked, with a deep hypural plate, wide dorsal scute, and a prominent pre-caudal scaly keel on the caudal peduncle.

#### 3.2.2. Cranium

The skull is perfectly articulated and mostly complete, aside from the skull roof and dorsal region of the operculum, which are inaccessible due to the specimen's ventrolateral exposure on the slab (Figure 3). Taphonomic compaction has distorted the orbital region, causing the antorbital, nasals, and ossicles of the sclerotic ring to fragment. Some thinner, more ductile bones of the operculum and ventral gill skeleton display a 'shrink-wrapped' texture, whereby their compression over more resistant bones preserves the underlying topography of the internal bones, which are not normally exposed. This permits examination of some of the internal anatomy in these regions.

**Table 1.** Morphometric values of *Germanostomus pectopteri* gen. et sp nov., and other basal asthenocormines. Values for *Saurostomus esocinus* are from Reference [24]. All measurements are in millimeters. BL = body length; CFH = caudal fin height; HL = head length; MH = mandible height; ML = mandible length; OPH = operculum height; OPL = operculum length; PCFL = pectoral fin length; PCFW = pectoral fin width; SL = standard length; TL = total length.

| Specimen no. | Taxon | TL | SL | BL | CFH | HL | ML | MH | POL | OPL | PCFW | PCFL |
|---|---|---|---|---|---|---|---|---|---|---|---|---|
| SMNS 15815 | *Germanostomus pectopteri* gen. et sp. nov. (Holotype) | 1060 | 850 | 605 | 90 | 245 | 125 | 25 | 55 | 120 | 130 | 190 |
| SMNS 56344 | *Germanostomus pectopteri* gen. et sp. nov. | - | - | - | - | - | - | - | - | - | 85 | 140 |
| GPIT-PV-31531 | *Ohmdenia multidentata* (Holotype) | - | - | - | - | - | 598 | 95 | - | 165 | 100 | 335 |
| SMNS 51144 | *Saurostomus esocinus* (neotype) | 750 | 630 | 480 | 212 | 182 | 90 | 20 | 40 | 47 | 100 | 150 |
| SMNS 12576 | *Saurostomus esocinus* | 1260 | 1100 | 830 | 350 | 280 | 140 | 30 | 58 | 120 | 177 | - |
| SMNS 18189 | *Pachycormus macropterus* | 1005 | 816 | 575 | 263 | 250 | 142 | 31 | 65 | 110 | 104 | 222 |

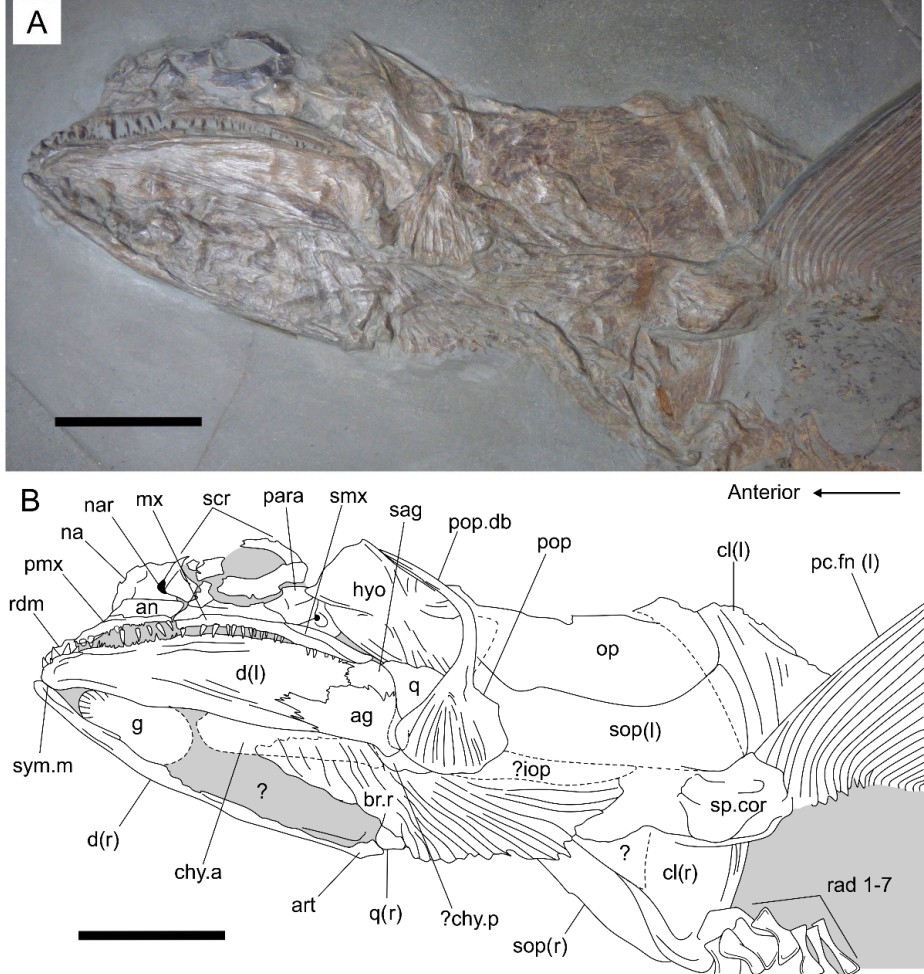

**Figure 3.** Skull and pectoral girdle of *Germanostomus pectopteri* gen. et sp. nov., holotype SMNS 15815: (**A**) skull in left lateroventral view; (**B**) annotated line drawing. ag, angular; an, antorbital; art, articular; br.r, branchiostegal rays; chy.a, anterior ceratohyal; chy.p, posterior ceratohyal; cl, cleithrum (l, left; r, right); d, dentary; g, gular; hyo, hyomandibula; ?iop, interopercle; mx, maxilla; na, nasal; nar, (?anterior) naris; op, opercle; para, parasphenoid; pc.fn, pectoral fin; pmx, premaxilla; pop, preopercle; pop.db, dorsal branch of the preopercle; q, quadrate; rad 1–7, radials 1–7; rdm, rosterodermethmoid; sag, surangular; scr, sclerotic ring; smx, supramaxilla; sop, subopercle; sp.cor, scapulocoracoid; sym.m, mandibular symphysis. Scale bars = 50 mm.

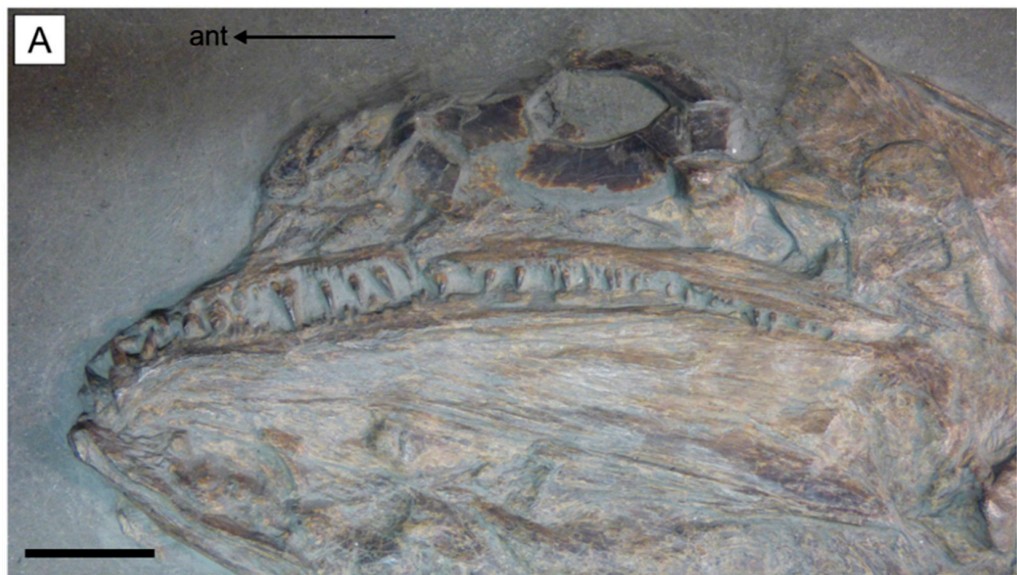

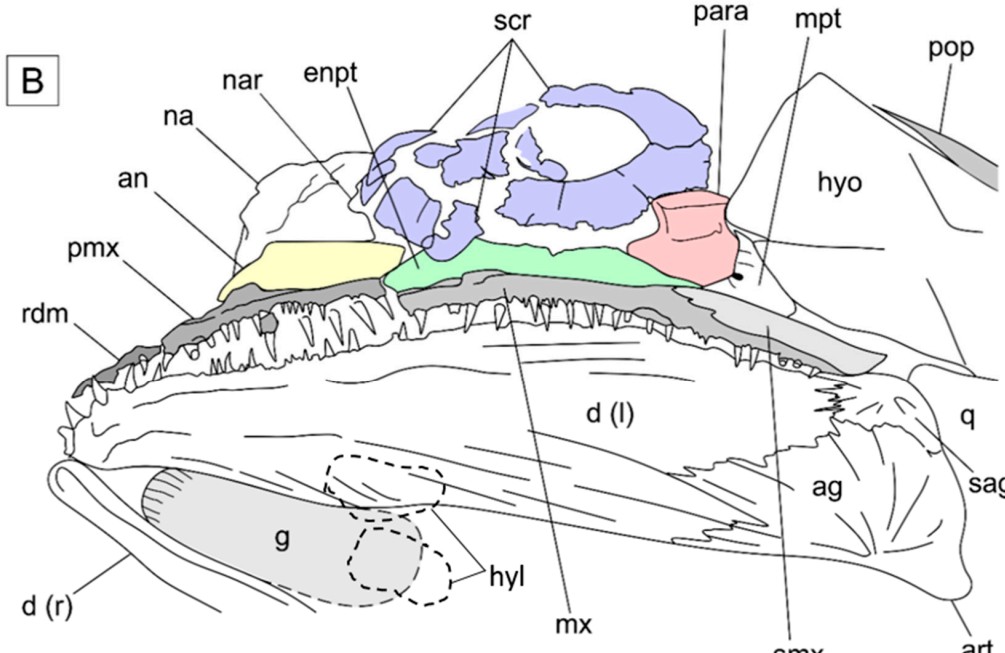

**Figure 4.** Jaws and orbital region of *Germanostomus pectopteri* gen. et sp. nov.: (**A**) photograph of the holotype SMNS 15815 in left lateral view; (**B**) interpretive line drawing with false color added to help differentiate bone boundaries. ag, angular; an, antorbital; art, articular; d, dentary (l, left; r, right); enpt, endopterygoid; g, gular; hyl, hypohyals; hyo, hyomandibula; mpt, metapterygoid; mx, maxilla; na, nasal; para, posterior region of the parasphenoid; pop, preopercle; pmx, premaxilla; q, quadrate; rdm, rostrodermethmoid; sag, surangular; scr, sclerotic ring; smx, supramaxilla. Scale bar = 20 mm.

*Orbital region*. The orbital region is incompletely preserved in the holotype, with the few exposed elements highly fragmented as a result of compaction (Figure 4). The superficially rectangular antorbital is incomplete anteriorly and decreases in height towards the orbit, where it borders the anteroventral margin of the sclerotic ring. A similar placement and morphology is also present in *Saurostomus esocinus* [24] (Figure 4). The external surface of the bone has been roughly prepared, which has destroyed any trace of the ethmoid commissure. A trapezoidal bone situated dorsal to the antorbital and anterior to the sclerotic ring is interpreted as remnants of the left nasal; unfortunately, the posterior border is too

poorly preserved to assess the presence of a concave notch, as in *S. esocinus* [24]. The fragmented scleral ossicles are wide and flat; however, the external bone surfaces are smooth, as opposed to ornamented as in *S. esocinus*. The original shape of the sclerotic ring (ovate in *S. esocinus* [24], or spherical in more basal forms, e.g., *Pachycormus macropterus* [23,27,28]) is uncertain due to the fragmentation of the scleral ossicles in the holotype. The infraorbitals, postorbitals, and suborbitals are all absent in *Germanostomus pectopteri* gen. et sp. nov., which is consistent with their shared absence in both *S. esocinus* and the suspension-feeding clade [1,17,20,24]. Their absence in *Germanostomus*, as in *Saurostomus*, externally exposes the hyomandibula as well as the posterior region of the palate and suspensory apparatus (see below). The bones that would be covered by the suborbitals in other taxa (e.g., the hyomandibula, metapterygoid, and quadrate; see Reference [27]) are well prepared in SMNS 15815, which rules out either rough preparation or specimen damage as possible explanations for the absence of the suborbitals. Furthermore, the holotype was prepared from its lower surface; therefore, it would be illogical to suggest that the bones dislocated and floated away before final burial.

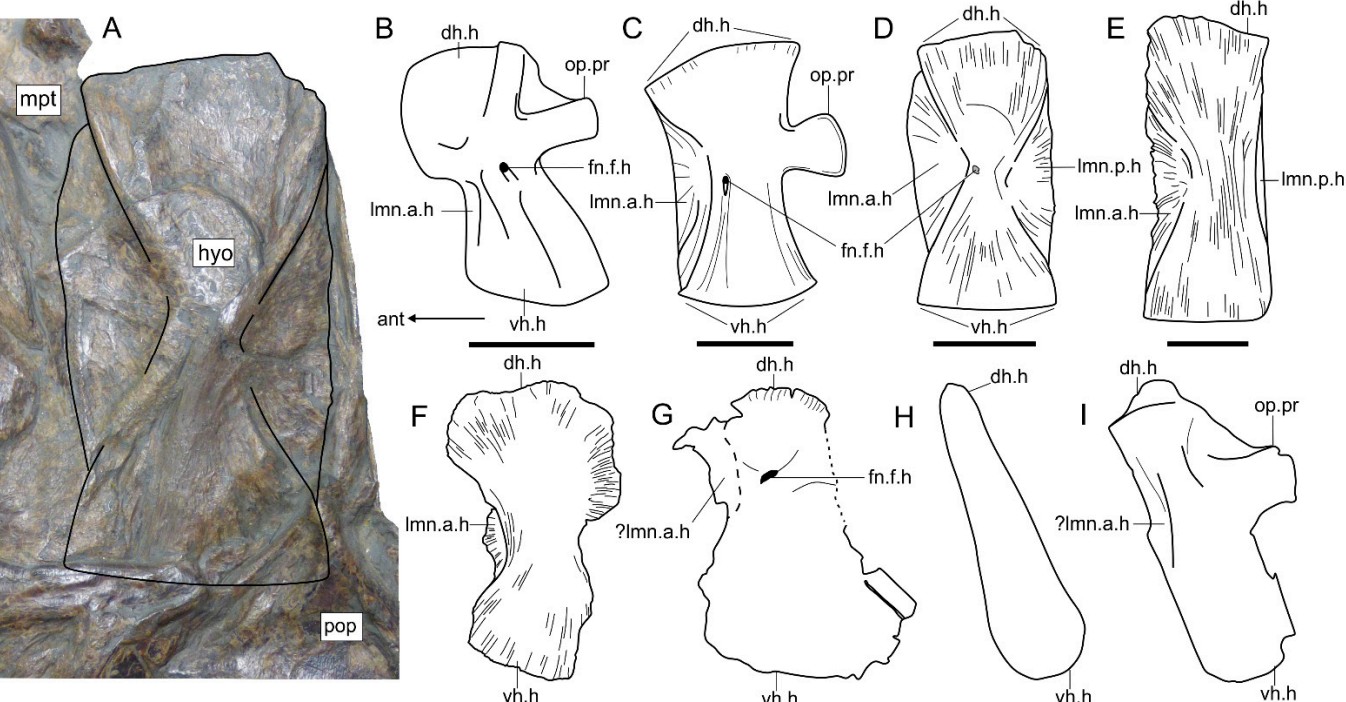

**Figure 5.** Comparative hyomandibular morphology in pachycormids: (**A**) left hyomandibula of SMNS 15815, *Germanostomus pectopteri* gen. et sp. nov., in lateral view; (**B**) *Pachycormus macropterus* (redrawn from Reference [27]); (**C**) *Saurostomus esocinus* (redrawn from Reference [24]); (**D**) *Germanostomus pectopteri* gen. et sp. nov., interpretive drawing of (**A**); (**E**) *Ohmdenia multidentata*, holotype (GPIT-PV-31531); (**F**) *Leedsichthys problematicus*, based on NHMUK PV P 1823 (mirrored); (**G**) *Martillichthys renwickae*, holotype (NHMUK PV P 61563, drawing based on Reference [20]); (**H**) *Rhinconichthys* sp. (redrawn from Reference [17]); (**I**) *Bonnerichthys gladius*, holotype (KUVP 60692, redrawn from Reference [5]). dh.h, dorsal head of the hyomandibula; fn.f.h, foramen for the facial nerve of the hyomandibula; hyo, hyomandibula; lmn.a.h, medial-anterior lamina of the hyomandibula; lmn.p.h, medial-posterior lamina for the hyomandibula; mpt, metapterygoid; op.pr, opercular process of the hyomandibula; pop, preopercle; vh.h, ventral head of the hyomandibula. Scale bars = 30 (**A–D**), 50 (**E,G–I**), and 100 mm (**F**).

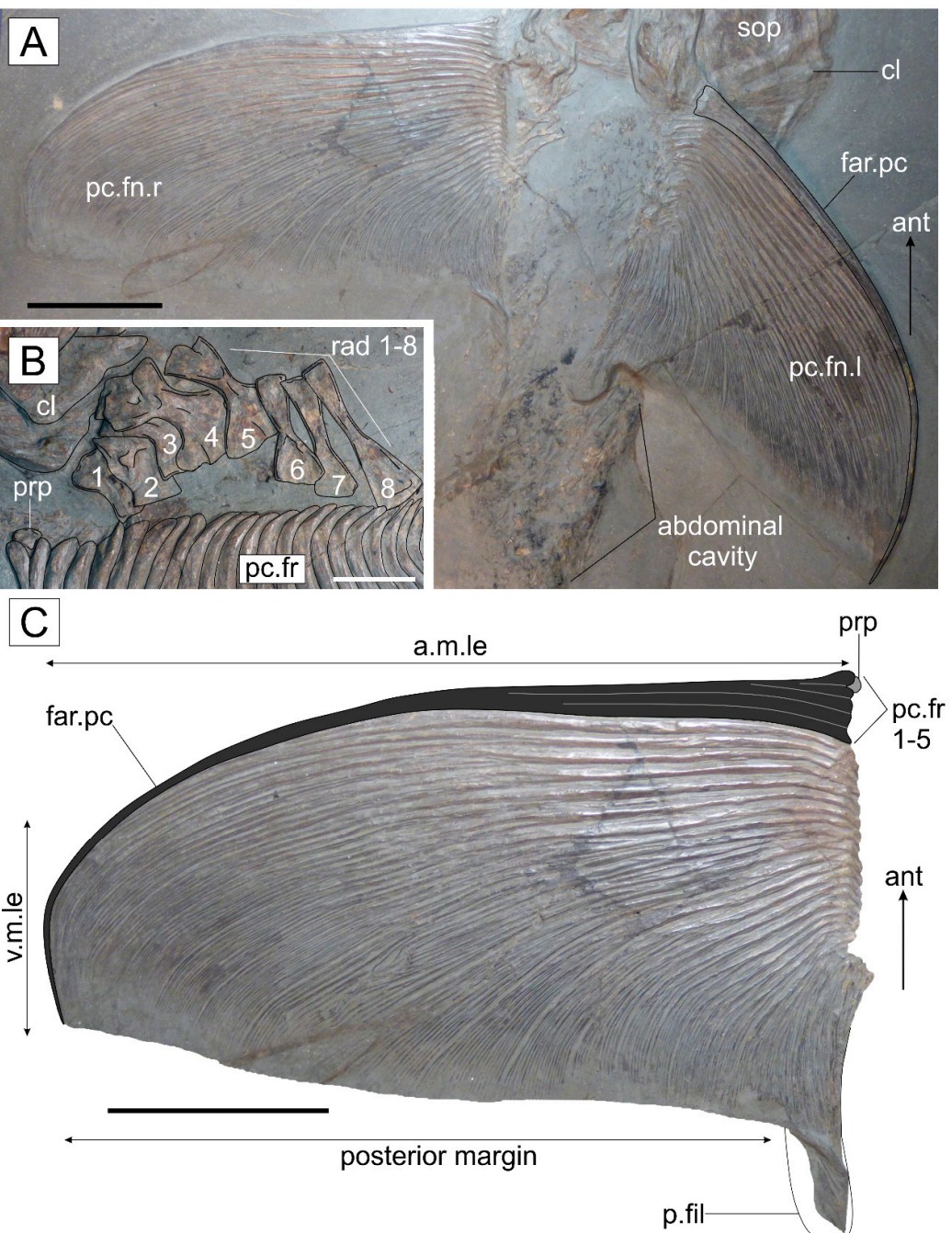

**Figure 6.** *Germanostomus pectopteri* gen. et sp. nov., endochondral pectoral girdle and pectoral fins of the holotype (SMNS 15815): (**A**) pectoral girdle showing pectoral fins prepared in ventral view; (**B**) right endochondral pectoral girdle exposing the articulated radials and small propterygium; (**C**) right pectoral fin showing the fused anterior spine of the leading edge. The posterior-most lepidotrichia of the posterior fillet have folded over during burial; the outline indicates their original extent. a.m.le, anterior margin of the leading edge; ant, anterior direction; cl, cleithrum; far.pc, fused anterior spine of the pectoral fin; p.fil, posterior fillet; pc.fn, pectoral fin (l, left; r, right); pc.fr, pectoral fin rays; prp, propterygium; rad 1–8, radials 1 to 8; v.m.le, distal margin of the fin. Scale bars = 50 (**A,C**) and 10 mm (**B**).

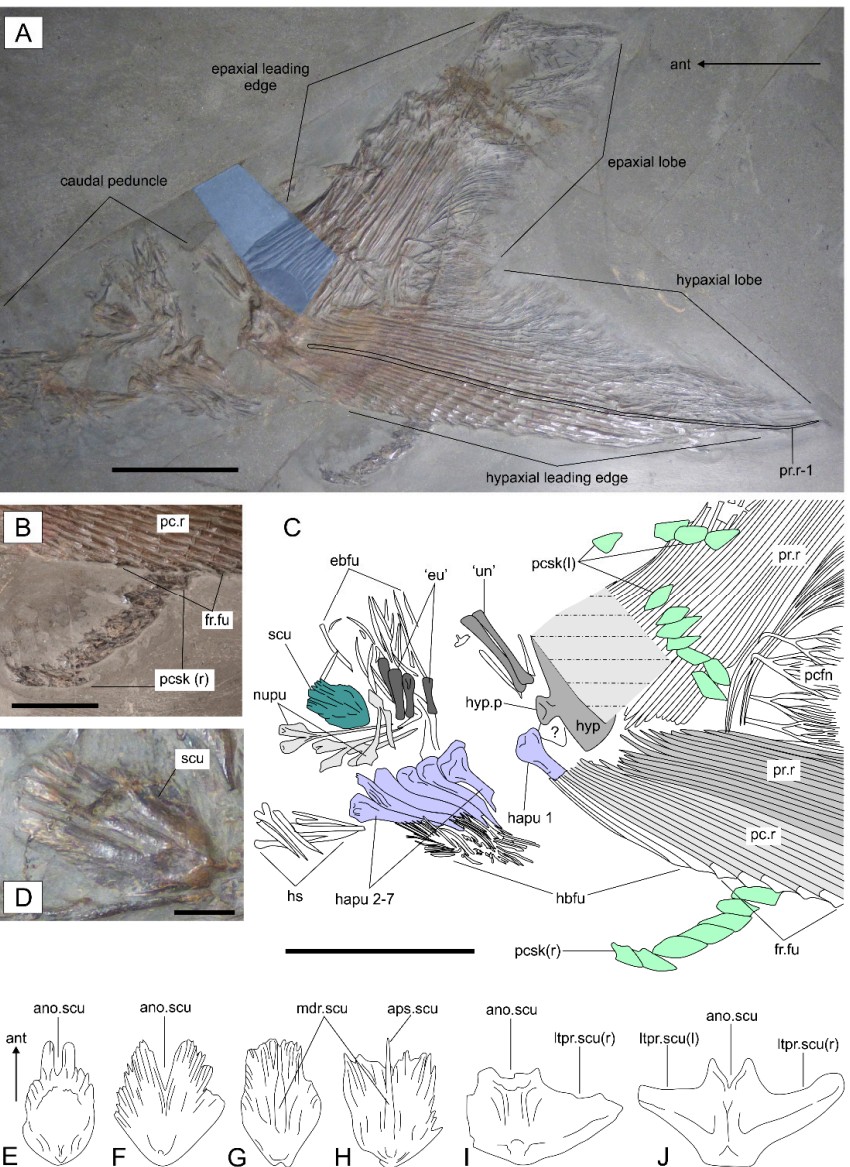

**Figure 7.** Caudal region of *Germanostomus pectopteri* gen. et sp. nov.: (**A**) holotype (SMNS 15815) in left lateral view. Areas of the specimen that have been reconstructed are shaded in blue. (**B**) Details of the right pre-caudal scaly keel and hypaxial fringing fulcra. (**C**) Line drawing of the caudal endoskeleton with shading and false colours indicating different pre-hypural elements and fin ray types. (**D**) Basal dorsal scute in external view showing characteristic anterior ornamentation and the medial dorsal ridge. Comparative line drawings of the basal dorsal scutes of closely related pachycormid taxa (**E–J**). (**E**) *Saurostomus esocinus,* neotype, immature individual (SMNS 51144); (**F**) *Saurostomus esocinus,* mature individual (SMNS 56982); (**G**) *Germanostomus pectopteri* gen. et sp. nov., holotype (SMNS 15815); (**H**) *Ohmdenia multidentata,* holotype (GPIT-PV-31531); (**I**) *Martillichthys renwickae,* holotype (NHMUK PV P 61563); (**J**) *Asthenocormus titanius* (L.1309; drawn from figure 7.4 in Reference [1]). ano.scu, anterior notch of the basal dorsal scute; aps.scu, anteriorly protruding spine of the dorsal scute; ebfu, epaxial basal fulcra; 'eu', epural-like elements; fr.fu, fringing fulcra; hapu 1–7, haemal arches/spines of the preural vertebrae; hs, haemal spines; hyp, hypural plate; hyp.p, hypural process; ltpr.scu, lateral processes of the basal dorsal scute (l, left; r, right); mdr.scu, medial dorsal ridge of the basal dorsal scute; nupu, neural arches/spines of the preural vertebrae; pcfn, posterior caudal fans; pc.r, procurrent fin rays; pcsk, pre-caudal scaly keel (l, left; r, right); pr.r, principal fin rays; pc.r-1, first principal caudal fin ray; scu, basal dorsal scute; 'un', uroneural-like elements. Scale bars = 50 (**A**,**C**); 20 (**B**); and 5 mm (**D–J**).

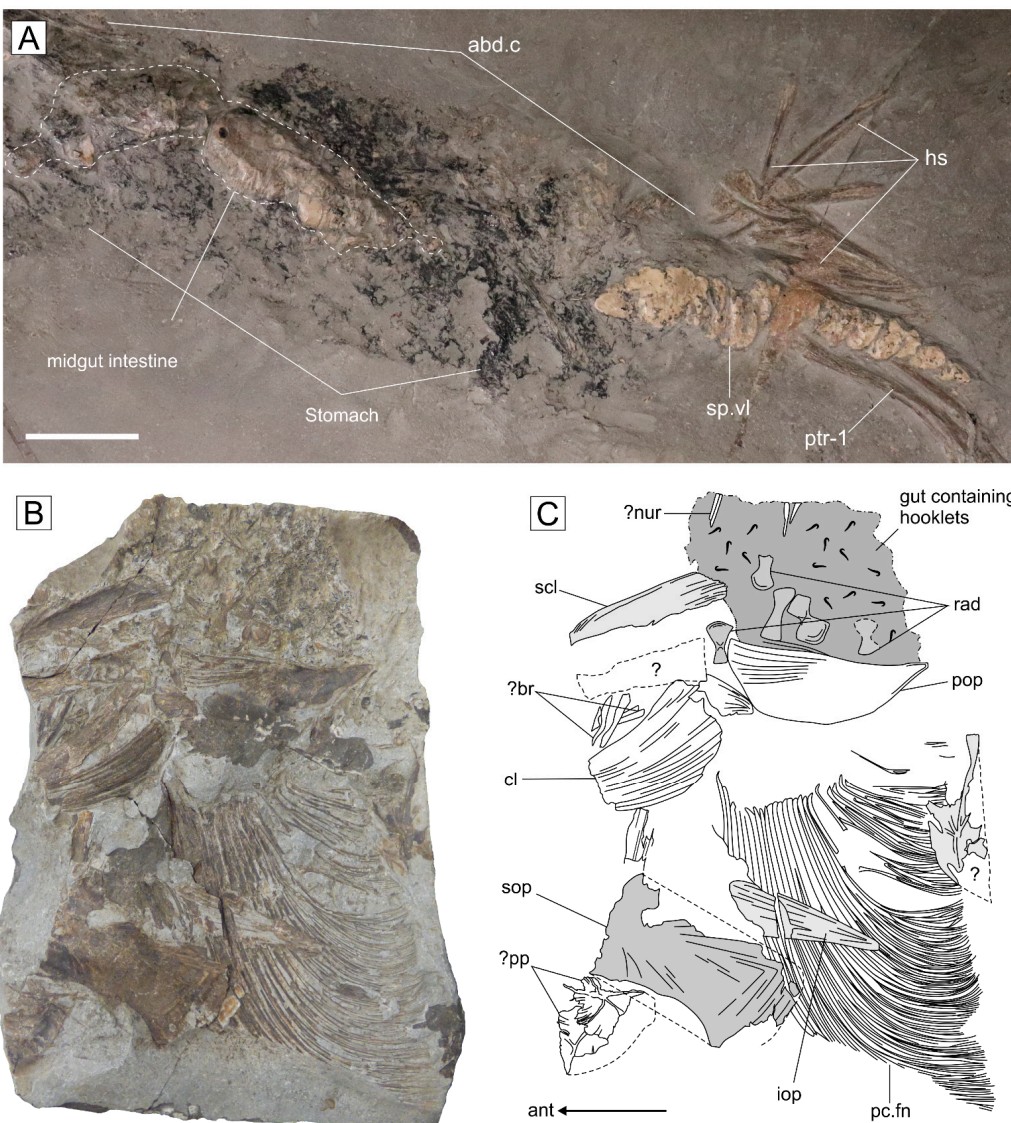

**Figure 8.** Gastric contents and alimentary canal in *Germanostomus pectopteri* gen. et sp. nov.: (**A**) Abdominal cavity of SMNS 15815 (holotype) preserving the phosphatized midgut intestine and the spiral valve of the alimentary canal. The black cloud infilling the abdominal cavity likely represents the stomach periphery. (**B,C**) Photograph (**B**) and interpretive line drawing (**C**) of the paratype concretion (SMNS 56344), preserving the left pectoral fin in addition to disarticulated elements of the operculum, pectoral girdle, neural spines, and possibly elements of the skull roof. Remnants of the gut situated dorsal to the pectoral fin preserve abundant coleoid hooklets. abd.c, abdominal cavity; ?br, branchiostegal rays; cl, cleithrum; hs, haemal spines; iop, interopercle; ?nur, neurals; pc.fn, pectoral fin; pop, preopercle; ?pp, postparietals; ptr-1, first anal pterygiophore; rad, radials; scl, supracleithrum; sop, subopercle. Scale bars = 50 mm.

*Suspensory apparatus and palate.* The quadrate is triangular, with a widened dorsal border and a powerful, well-ossified ventral condyle for articulation with the quadratomandibular joint of the lower jaw. The posterior face of the quadrate is tightly sutured to the ventroanterior margin of the preopercle, as per the condition in mature specimens of *S. esocinus* and *Ohmdenia*. The wide dorsal margin of the quadrate contacts and slightly overlies the ventral surface of the metapterygoid. The symplectic is not exposed.

The hyomandibula is symmetrically rectangular, tall, and displays gentle parallel waisting on both sides of the midline. The posterior margin of the hyomandibula is straight

and flat and is lacking an opercular process (Figure 5A,D). A narrow medioposterior lamina forms a buttress between the dorsal and ventral heads of the shaft—it extends dorsally along the dorso-posterior leading edge—occupying the position held by the opercular process in *S. esocinus*. Both the dorsal and ventral heads of the bone are equal in width, and they are slightly convex, with gently sloping anterior and posterior edges that contact one another close to the midline—forming a superficial hourglass contour. The medioanterior lamina extends the entire height of the bone, similar to its medioposterior counterpart, but slightly wider. The foramen for the facial nerve of the hyomandibula (fn.f.h) cannot be located on the holotype. The hyomandibula of *Saurostomus esocinus* is more strongly waisted, with sharply expanded, hatched-shaped distal heads. *S. esocinus* lacks a medioposterior lamina but instead retains a fully developed opercular process [24] (Figure 5C).

The palate is very poorly exposed in the skull, where it is naturally overlain by the upper jaw and sclerotic ring. A thin rectangular ossification with a straight dorsal margin partially exposed dorsal to the upper jaw and between fragments of the sclerotic ring is interpreted as the endopterygoid based on its comparative position in the skulls of *Pachycormus* [27] and *Saurostomus* [24]. The metapterygoid is partially exposed between the supramaxilla and hyomandibula. The dorsal surface of this bone is strongly concave, with a distinct notch representing the groove for the mandibular branch of the trigeminal nerve (*sensu* Mainwaring [27]) situated on the midline.

*Upper jaw*. The upper jaw, comprising the premaxilla, maxilla, and supramaxilla, is elongated well behind the orbit, and is mostly straight, aside from a gentle ventral curvature at the maxilla–supramaxilla contact (Figure 4). The rostrodermethmoid, which contributes to the anterior border of the upper jaw in all pachycormids [1,24,27,28,80], has shifted behind the lower jaw during compaction and, thus, is not usefully exposed in the new species. Only the left side of the lateral margin of the rostrodermethmoid is partially exposed, where it preserves at least four broken tooth bases, confirming the rostrodermethmoid to be dentigerous, as in all members of Pachycormiformes outside of the derived suspension-feeders (e.g., [1,5,16,20]).

The premaxilla is shallow and elongate, contributing approximately 33% (1/3) of the total length of the upper jaw. The premaxilla is relatively shorter in *S. esocinus*, where it contributes ≤17% of the total length of the upper jaw [24]. Dorsally, the premaxilla contacts almost the entire ventral margin of the antorbital, with only the posterior-most end of the antorbital contacting the maxilla. In *Saurostomus esocinus*, the ventral margin of the antorbital is almost entirely contacted by the maxilla, with the premaxilla restricted to the anterior-most margin of the bone. Eight large principal premaxillary teeth are present in *Germanostomus pectopteri* gen. et sp. nov., which is approximately double the maximum count of four in *S. esocinus* [24]. Additionally, the premaxilla of *G. pectopteri* gen et sp. nov., is not laterally expanded to hold additional rows of teeth, as seen in *S. esocinus*; rather, only a single principal row of well-spaced, stout conical teeth is present in the new species. The eruption of additional teeth on the coronoids and premaxilla is attributed to increased osteological maturity for *S. esocinus*, first seen in individuals with total lengths exceeding 750 mm [24]. The large body size of the holotype of *Germanostomus* (1060 mm TL) does not support osteological immaturity as the cause of this variation between *Germanostomus pectopteri* gen. et sp. nov. and *S. esocinus*.

The free maxilla is only sutured anteriorly to the immobile premaxilla, and extends well behind the orbit, where it terminates immediately before the hyomandibula. The elongated maxilla is superficially lenticular, slightly triangular in cross-section, and is of equal thickness to the premaxilla, but begins to sharply narrow to a terminal point below the contact with the supramaxilla. The maxillary teeth are stout, conical, and well spaced along the principal tooth row. Unlike *Saurostomus esocinus*, the teeth are straight as opposed to curved, and are devoid of any ornamentation or vertical grooves at the bases. The teeth in the marginal row are dramatically smaller in comparison to those in the principal row, and are composed of simple lanceolate crowns that extend continuously from the premaxilla and terminate just below the anterior extremity of the supramaxilla.

The supramaxilla is edentulous, lenticular in shape, approximately twice as wide as tall, and fused to the dorsoposterior slope of the maxilla. Posteriorly, the supramaxilla extends further back than the maxilla, forming the distal extremity of the upper jaw.

*Lower Jaw.* The mandible, composed externally of the dentary, angular, surangular and articular, is elongated and gracile, accounting for approximately half (50%) of the total skull length and is articulated well behind the orbit. The lower jaw measures 125 mm in length and 25 mm in depth, with an approximate length-to-depth aspect ratio of 5:1. The aspect ratio is therefore greater than the 4:1 ratio in *Saurostomus esocinus* [24], but is only half that of the more deeply nested asthenocormine *Ohmdenia multidentata* (jaw ratio = 8:1; Table 1).

The dentary is elongated and moderately shallower than in *S. esocinus*, with a slightly undulated dentigerous surface and an irregular ventral margin of the ramus which increases in depth towards the posterior. The rami are relatively straight but exhibit a strong inward curvature towards the mandibular symphysis, which is only acutely expanded in this region in contrast to *S. esocinus*. Ornamentation of the mandible is consistent with that of both *Saurostomus* and *Ohmdenia* [24] (Figures 3 and 4). Teeth on the dentary are well spaced, conical with narrow unornamented bases, and all display straight crowns (as opposed to curved in *S. esocinus*). The flanking teeth on the marginal row are significantly smaller and extend the entire length of the dentigerous margin. The anteriormost teeth close to the symphysis are inclined slightly forwards, although they are not as steeply inclined anteriorly ($\approx$180° to the long axis of the dentary) as they are in mature examples of *S. esocinus* ($\geq$1000 mm TL) as well as in *Ohmdenia multidentata* (Figure S9).

The placement and morphology of the surangular and angular are consistent with those in *S. esocinus*, although by contrast these are proportionately smaller relative to the length of the mandible on account of the increased elongation of the dentary in the new species. Interestingly, the increased elongation of the mandible is affected only by the elongation of the dentary; the surangular and angular remain conservative. This allometric asymmetry is observed to a greater degree in the lower jaw of *Ohmdenia multidentata* [16,31] (Figure S9).

The articular is only identified by its underlying topography below the surangular, where it is shown to be superficially 'trigger shaped', with a deep articular cotyle for the quadratomandibular joint at the posteroventral extremity of the lower jaw. The coronoids and prearticular are not well exposed, but there is no indication of a coronoid process. The mandibular sensory canal, which in pachycormids originates close to the oral margin of the dentary and extends ventrally through the lower half of the angular [15,26,27,33,81], is not externally discernible in the lower jaw of SMNS 15815.

*Operculum.* The operculum contributes to just under half of the head length and is composed of the preopercle, opercle, subopercle, and interopercle. The preopercle is similar to that of *S. esocinus* in that it is deep and slender with an expanded ventro-posterior fan. The splint-like dorsal portion of the preopercle displays a strong dorsoanterior curvature, forming a semi-lunate topography of the bone. The ventroposterior fan is drastically narrower and shorter than in *S. esocinus*, although both species share similar ornamentation composed of evenly spaced, dorsoventral ridges (Figure 3). The curvature of the dorsal portion is less pronounced and sigmoidal in *S. esocinus*, with the ventroposterior fan extending roughly half the height of the bone.

The incomplete opercle contributes the largest portion of the operculum and is situated dorsal to the subopercle and dorsoposterior to the preopercle. The dorsal portion of the opercle is incomplete; however, it appears to share the same trapezoidal morphology as *Saurostomus esocinus* with its line of the maximum width situated closer to the ventral margin than the midpoint. Externally, the bone is lightly ornamented with fine ridges, which radiate outwards towards the posterior margin from the anterior midpoint.

The subopercle is rectangular with a convex posterior border and is slightly overlapped dorsally by the ventral margin of the opercle. Externally, the subopercle bears the same ornamentation as the opercle, and both bones are of an equal anteroposterior length, with the hypothetical line of maximum width situated along the dorsal margin of the subopercle.

The interopercle is difficult to differentiate in the holotype due to the 'shrink-wrapped' compaction distortion in the ventro-posterior region of the skull. However, due to the articulation with the operculum in SMNS 15815, it can be confidently identified as an interopercle as opposed to a branchiopercle, which articulates with the branchiostegal rays in taxa where it is present (e.g., †*Watsonulus*, *Amia*: [6,68]). The interopercle is better preserved in SMNS 56344, where it is fully exposed in isolation from the rest of the skull (Figure 8B). The interopercle is the smallest component of the operculum, with a length only comparable to one-third that of the subopercle. The elongated interopercle is semicircular with a straight dorsal margin and a deep, convex curvature along its ventral margin. It is similarly ornamented to the preopercle, with well-spaced, longitudinal ridges which are localized to the dorsoposterior region of the bone. By contrast, the interopercle of *S. esocinus* is noticeably smaller, topographically sigmoidal, and does not display any recognizable ornamentation.

*Ventral gill skeleton and hyoid arch.* Most of the ventral gill skeleton and hyoid arch, with the exception to the hyomandibula, is poorly exposed due the overlying branchiostegal rays obscuring much of their morphology. The gular plate occupies just over a third of the mandibular length and is strongly ovate with equally wide anterior and posterior rounded margins. Although imperfectly preserved, the anterior margin retains evidence of a serrated marginal ornamentation akin to that of the gular plate of *Saurostomus esocinus*. It is uncertain if the gular displays the sinistral/dextral asymmetry seen in *S. esocinus*. A pair of semi-lunate ossifications below the gular likely represent the paired hypohyals (Figure 4B) based on their similar placement in *S. esocinus* [24] (Figure 7B,C), *Pachycormus* [27,28], and *Martillichthys* [13,20]. Likewise, the outline of a larger rectangular bone that articulates posterior to the hypohyal likely corresponds to the left anterior ceratohyal (Figure 3).

The ventral gill skeleton is entirely hidden; however, like the hyoid arch, the 'shrink-wrapped' compaction texture of the overlying tissues has revealed the topography of some of these elements. At least four elongated stem-shaped bones extending from the region of the gular to just behind the lower jaw are interpreted as ceratohyals, although their exact number, extent, and region of contact with the hypobranchials is uncertain. The anterior region of the gill skeleton, comprising the hypobranchials and interhyal, is entirely hidden in the holotype.

At least 17 pairs of branchiostegal rays are articulated in life position along the ventral margins of the anterior and posterior ceratohyals of the hyoid arch. They are elongated and curved posteriorly, with the last few extending well behind the lower jaw and terminating just below the preopercle-subopercle contact. Their general morphology and arrangement are very similar to those of *Saurostomus esocinus* [24,33]. There is no evidence for a branchiopercle, which is sometimes present in more primitive actinopterygians (e.g., †*Watsonulus*, †*Ophiopsiella*, *Amia* [6,68,82,83]).

### 3.2.3. Postcranium

*Vertebral column.* The postcranial axial skeleton, including most notably the ossified neural and haemal arches and spines of the vertebral column, are highly disrupted, with most elements either lost or displaced in the holotype. Inferred dorsal side-up arrival of the carcass on the sea floor most plausibly explains this unusual preservation, wherein anterior and ventral regions of the carcass were immediately buried in soft clay with the pectoral fins acting as a hydrofoil upon landing, while dorsal elements were exposed on the surface, leading to a higher level of disruption and loss from prolonged exposure.

The anterior vertebral column is poorly represented, with the supraneurals, anterior neural arches, and abdominal ribs either absent due to the taphonomic disruption or obscured by the dense gastric contents. The abdominal neural arches are mostly all lost, al-though a few elements that may represent neural arches are partially exposed under the intestinal tract of the abdominal cavity. A few unpaired haemal spines are preserved in the anal region of the body. They are morphologically similar to those of *Saurostomus esocinus* and *Ohmdenia*, in that they are elongated but narrow with short, spatulate haemal

arches lacking articular processes. Due to the severe disruption and incompleteness of the vertebral column, it is not possible to either count the original number of vertebral segments or to comment on axial regionalization.

At least nineteen neural arches are preserved, with the remaining elements mostly concentrated between the anal fin and caudal peduncle. Those set most anteriorly are highly disrupted, while the posterior neural arches generally maintained their original articulation above the haemals. Neural arches closest to the anterior are incredibly deep and narrow with expanded spatulate bases; although they become significantly shorter and wider the closer they are to the caudal fin. Neural arches are further differentiated from their abaxial counterparts by the presence of a poorly developed posterior zygapophysis on the arches.

*Pectoral girdle*. The large cleithrum of *Germanostomus pectopteri* gen. et sp. nov. is similar to that of both *Saurostomus* and *Ohmdenia* [16,24,33] in that it is superficially crescent-shaped with a wide dorsal margin and a blade-like anteroventral limb, providing powerful articulation between the pectoral fin and neurocranium. The ventral limb of the cleithrum extends well below the operculum, and is characterized by a distinctive thickening of the bone along the posteroventral margin. The dorsal portion of the cleithrum is noticeably wider than the ventral portion, although the uppermost section of the cleithrum and the articulating supracleithrum are missing in the holotype. SMNS 56344 preserves a disarticulated portion of the pectoral girdle, including a section of the cleithrum and a thin and elongated lenticular bone that may represent a portion of the supracleithrum (Figure 8B); however, this bone is broken distally, casting doubt on its identity. The dorsal and ventral postcleithra, which usually articulate with the posterior face of the cleithrum, are hidden beneath the overturned left pectoral fin in the holotype (Figure 6A).

The scapulocoracoid is very poorly exposed in the holotype, where it is mostly overlain by the cleithrum. A small region of this compound element is visible between the ventral border of the right cleithrum and the first radial; it is gently globose with a faint trisecting ridge on the external surface. Unfortunately, the bone is not well enough exposed to better describe or usefully compare with *S. esocinus*.

Eight proximal radials are perfectly aligned in their original position on the right pectoral girdle, where they articulate dorsally with the scapulocoracoid and ventrally with the principal rays of the pectoral fin (Figure 6B). Radial 1 (anterior) is noticeably shorter than the successive radials and is superficially rectangular with a wide but shallow ovate proximal head and a poorly defined medial waist. The posterior margin of radial 1 has slipped downwards behind the pectoral fin fays. Radial 2 is slightly taller by comparison, with a wider proximal and distal head that are medially separated by an acutely defined medial waist. Radials 3–8 are of a uniform height (19 mm), although their morphology changes gradually from anterior to posterior. Radials 3–4 are the widest by comparison, with strongly concave lateral waists forming equally larger proximal and distal convex heads for powerful articulation between the scapulacoracoid and pectoral fin rays. In radial 5, the lateral bone margins are more obtusely concave, with a large fan-shaped distal head that is almost twice as wide as the proximal head, which is comparatively narrower and spatulate. Radials 6–8 are morphologically dissimilar from their predecessors due to their increasingly narrow distal heads and extremely obtusely concave to vertically straight lateral bone profiles. The distal heads of the last three radials all possess a weakly spatulate morphology that becomes increasingly narrow towards the final radial (radial 8). The proximal heads are only gently expanded in relation to the medial shaft, and they vary from their predecessors (radials 1–5) by possessing flat, as opposed to convex, proximal margins. Collectively, the radials extend the entire width of the pectoral fin, with radial 8 articulating solely with the final few lepidotrichia which form the small posterior fillet (see pectoral fin).

The propterygium is imperfectly preserved in articulation with the expanded proximal base of the first lepidotrichium of the right pectoral fin (Figure 6B,C). The bone is small

(2 × 2 mm) and subspherical, with a slightly inflated dorsal portion, similar to the inverted 'egg-shaped'-morphology reported in S*aurostomus esocinus* [24].

*Pectoral fin.* The pectoral fin is wide and deep, making it the most salient diagnostic feature in *Germanostomus pectopteri* gen. et sp. nov. The right pectoral fin is most perfectly preserved in the holotype, measuring 190 mm in length and 130 mm in basal width, with a fin length-to-width ratio of 1.5: 1 (Figure 6C). The left pectoral fin is also complete and perfectly articulated, but has been slightly folded backwards from its life position relative to the right pectoral fin. The anterior leading edge of both pectoral fins is obtusely curved towards the posterior, with each distally bifurcating lepidotrichium terminating along the same transverse plane as the anterior fin ray, creating a perfectly straight posterior margin and giving the fin an inverted 'D'-shaped topography. The distal curvature of the leading anterior fin ray encapsulates the entirety of the distal margin of the pectoral fin, setting the distal termination at approximately 90° posterior to the vertebral axis, precluding the development of the whip-like extension seen in some specimens of *Saurostomus esocinus* [24,65,84]. This distinctive pectoral fin morphology is also present in the paratype (Figure 8B). The pectoral fin is composed of approximately 28 lepidotrichia, including those associated with the posterior fillet. The anterior fin ray is a compound element formed from the proximal fusion of the first three lepidotrichia into a dense anterior rod, which is strongly curved ventral-posteriorly to support the distinct curvature in the pectoral fin. The following 25 fin rays each decrease in length towards the posterior, and they are distally bifurcated asymmetrically and free from joints towards the transversely straight posterior margin. Proximally, the lepidotrichia are evenly aligned longitudinally in a straight line (as opposed to sigmoidal in *Saurostomus esocinus*) and are articulated with the propterygium (lepidotrichia 1–2) and the ventral margin of radials 1–7 (lepidotrichia 3–23). The final five lepidotrichia that form the posterior fillet are not articulated with the pectoral girdle and are instead attached freely to the posterior surface of lepidotrichium 23 (Figure 6A,C). The minute posterior fillet, which measures approximately 25 (anteroposterior width) × 8 mm (proximodistal length) comprises a shallow lobe-like projection at the dorsal-posterior corner of the fin, formed entirely of the final few lepidotrichia. These final lepidotrichia are well bifurcated with their primary split occurring close to the proximal bases of the fin ray (Figures 6C and 8B).

Anterior ossifications along the leading edge, which do not extend the entire length of the fin, are usually classified as basal fulcra (see References [64,69]) and, thus, the first two fins rays in the pectoral fin may be identified as such features. Similar structures, which also contribute to the fused anterior fin ray of *Saurostomus* (see [24] figure 11) were recently suggested to be possible basal fulcra [6]. Given that these anterior elements are laterally and distally fused together, they cannot be classified as basal fulcra under the current definition [64,69]. Furthermore, the first of these elements directly articulates with the propterygium in *Germanostomus pectopteri* gen. et sp. nov., further supporting our identification of these anterior elements as fused lepidotrichia (see Reference [85] for details on the contacts within the actinopterygian endochondral pectoral girdle).

The precise termination of the leading fin rays adjacent to all successive rays, with the exception of the posterior fillet, is unusual and displays high morphological divergence from the typical sickle to falciform pectoral fin morphologies of *Saurostomus esocinus* ([24]: figure 11) and more basal pachycormids (e.g., *Euthynotus* and *Pachycormus*; [1,23,33,65]). Morphologically, the pectoral fins of *Germanostomus pectopteri* gen. et sp. nov., are more closely comparable to *Ohmdenia multidentata* than they are with *Saurostomus esocinus* (see Discussion).

*Dorsal and anal fin.* The anal fin is wider (122 mm) than it is deep (90 mm) and is placed far back along the axial skeleton, terminating proximal to the caudal fin. Morphologically, it is falciform with a deep anterior blade which then shortens drastically into a wide but shallow tapering ventral edge. The first four anal lepidotrichia are the longest and form the anterior blade of the fin. Anteriorly, they are supported by shorter basal fulcra, numbering at least two; however, rough preparation along the leading edge has damaged some of these

elements meaning that their true count may be greater. The remaining 31 lepidotrichia are shorter, progressively decreasing in height towards the caudal fin. Each anal lepidotrichium demonstrates an unusual segmentation pattern, which is restricted to the medial region of each fin ray: the proximal region until the midpoint is unsegmented, whilst distally each ray segments up to four times before asymmetrically bifurcating in a similar fashion to the pectoral lepidotrichia. This unusually localized segmentation pattern is shared with *Saurostomus esocinus*; no other pachycormids are reported to display segmentation in their anal fin lepidotrichia. The anal fin endoskeleton is mostly absent due to the incompleteness in this region; however, a few anterior anal pterygiophores have dislocated forward into the abdominal region. Anal pterygiophores are thicker than the ribs or haemal spines and possess a more concave profile with a prominent dorsoanterior curvature. Their original number is unknown.

The dorsal fin and associated fin supports (pterygia) are entirely unknown, creating uncertainty regarding its placement along the axial skeleton.

*Caudal endoskeleton and caudal fin.* The caudal fin is homocercal, moderately forked and deep, with an angle of approximately 110° separating the upper (epaxial) and lower (hypaxial) lobes. Both lobes are equal in length, with the ventral lobe extending well below the anal fin, but the caudal fin lobes are shorter than the pectoral fins. Although moderately taphonomically disrupted, the caudal region of the holotype is mostly complete, aside from some disarticulation of the caudal endoskeleton. The caudal fin, including the hypural plate, have detached slightly from the vertebral column, with the dorsal region of the epaxial lobe exhibiting moderate disarticulation of the segmented fin rays. A small region of the caudal endoskeleton containing the anterior epaxial basal fulcra and the dorsal portion of the hypural plate is missing and has been replaced with sculpted plaster during preparation.

The caudal endoskeleton is incompletely preserved, with elements situated dorsal to the notochord either disarticulated or absent. Many of these elements are jumbled together, further complicating their identification and count. At least three 'uroneural-like' elements (see Reference [63]) are articulated dorsoanteriorly with the hypural plate. They are elongated with laterally compressed dorsal regions and a large, laterally interlocking club-like proximal base. In *Saurostomus esocinus*, the 'uroneural-like' elements number at least seven and maintain the same position relative to the hypural plate, but when perfectly articulated, they meet the preural neural spines anteroventrally and the epurals anterodorsally. However, the 'uroneural-like' elements in *S. esocinus* are more sigmoidal and are relatively taller in relation to the preural neural arches, with convex dorsal heads, rather than flat ones in the new species (Figure 7A,C).

Four 'epural-like' elements (see Reference [64] for issues with homology) are displaced anterior of to the 'uroneurals', where they maintain their ventral articulation with the anterior-most epaxial basal fulcra (Figure 7C). Their height is greater than their width, but they are half the height of the 'uroneural'-like elements (15 mm). They are laterally compressed and widened anteroposteriorly at their bases, where they articulate dorsally with the neural spines of the preural vertebrae. The morphology of the dorsal portion is unknown, as this region is overlapped by the articulating epaxial basal fulcra.

Preural neural arches/spines are incomplete, with as few as five preserved in the holotype. In anterior view, each paired arm of the neural arches is rhombic with wide convex ventral margins forming a narrow but tall neural canal. The arches are fully fused and articulatory zygapophyses are absent. In lateral view, the arches are tear-drop shaped whilst their narrow neural spines are elongate and straight. They articulate ventrally with the haemal arches of the preural vertebrae.

All seven preural haemal arches supporting the basal fulcra (hspu 1–7; *sensu* Reference [70]) are articulated and ventroposteriorly inclined, except hspu-1 (the parhypural), which is displaced along with the hypural region of the caudal fin (Figure 7C). Nine preural haemal arches are present in *Saurostomus esocinus* [24]. It is uncertain if the lower count in the new species is real or a consequence of incomplete preservation. Similar to *S. esocinus*,

the preural haemal spines of *Germanostomus pectopteri* gen. et sp. nov., are well ossified and are laterally expanded toward their dorsal bases to form club-like articular surfaces for rigid articulation with the notochord. In contrast to *Saurostomus esocinus*, which display a size and shape gradient in the preural haemal spines, those of *Germanostomus pectopteri* gen. et sp. nov., are of equal length (shorter than the hypural), each with a conservative convex topology to their haemal spine. The parhypural displays a dense convex dorsal head and is positioned anteroventral to the hypural plate where it is mostly obscured by the hypaxial caudal fin rays. Hspu 2–7 have detached from the caudal fin where they maintain their ventral articulation with the hypaxial basal fulcra (Figure 7C). The dorsal heads of hspu 2–3 are saddle-like, with a dorsal medial depression, whilst hspu 4–7 are more triangular with steeper lateral edges and a symmetrical convex dorsal margin (Figure 7C). Each of the haemal arches of the preural vertebrae articulate with several of the hypaxial basal fulcra, thus differentiating them from the regular haemal spines of the pre-caudal region of the axial skeleton (see Reference [62]).

The hypural plate comprises a large, laterally compressed fan with a height almost twice as great as its length. The number of hypurals incorporated into this structure is unknown in Pachycormidae [14,28,63]. The dorsoposterior region of the hypural plate is missing in the holotype, where it has been reconstructed with sculpted plaster. Located on the anteromedial margin is a median club-like articular hypural process flanked ventrally by a very small dome-like anterior process. In contrast, the anterior margins above and below the hypural process are perfectly straight, whereas they are undulating and irregular in *S. esocinus* (figure 12 of Reference [24]). In addition, the hypural process of *S. esocinus* is relatively much larger and globose with a more prominent anterior process than that of the new taxon. The ventral margin of the hypural plate is partially overlapped by the hypaxial procurrent fin rays (hypurostegy) – a condition shared with all pachycormids with well-preserved caudal fins (e.g., [63]). All procurrent rays in the caudal fin should articulate with the hypural plate in pachycormids [63,64], although this is not possible to test in the upper (epaxial) lobe due incomplete and misleading reconstruction in this region.

Basal fulcra are a series of numerous unpaired lanceolate ossifications aligned along the bases of both the epaxial and hypaxial lobes of the caudal fin, where they serve to reinforce the rigidity of the caudal fin and contribute the proximal-anterior portion of the leading edges. They are narrower than the fin rays and progressively decrease in size anteriorly, with those of the epaxial lobe buttressed anteriorly by the dorsal scute. The preserved epaxial basal fulcra are highly disrupted in the holotype, although a small cluster retains their original articulation with the dorsal portions of the displaced 'epurals' (Figure 7A,C). The base of the epaxial lobe (above the hypural plate), where the basal fulcra–procurrent ray transition is situated, is missing in the holotype. The hypaxial basal fulcra, numbering at least 13, are mostly aligned in their original life positions at the base of the ventral leading edge of the fin. Their morphology is identical to that of the upper lobe; they all articulate proximally with the ventral portion of the preural haemal arches (Figure 7C). In relation to size, the hypaxial basal fulcra gradually increase in length towards the posterior, where they contribute to approximately one-fifth of the leading edge of the hypaxial lobe before they are succeeded by the procurrent rays and fringing fulcra.

Fringing fulcra comprise a single series of scale-like terminal segments of the marginal lepidotrichia (fulcra), which contribute to roughly two-thirds of the leading edge of the hypaxial lobe in *Germanostomus pectopteri* gen. et sp. nov., but they are absent in the epaxial lobe. Fringing fulcra are formed by an expansion of the distal terminal segments of the procurrent rays, therefore conforming to 'pattern A' of Arratia [64,69]. Morphologically, the hypaxial fringing fulcra are very similar to those in *Saurostomus esocinus* (see [24] figures 10C and 12A); however, the distal terminations of the fulcra, which exhibit a distally expanded triangular topography, are less pronounced and notably smaller in *Germanostomus pectopteri* gen. et sp. nov. Fringing fulcra are not observed on the caudal fin of *Ohmdenia multidentata* (SC pers. obs.) and are absent in both *Martillichthys* (see description in Supplementary File S1; Figure S2) and *Asthenocormus* [1,63]. Cooper and Maxwell [24] recorded unusual

variability in the size of the hypaxial fringing fulcra relative to body size within *Saurostomus*, noting that they are significantly larger in individuals from the *tenuicostatum* Zone than those of the overlying *serpentinum* Zone. Their further reduced size in *Germanostomus pectopteri* gen. et sp. nov., from the middle *serpentium* Zone suggests an evolutionary reduction in the size of fringing fulcra within Asthenocorminae. This tendency may explain their absence in more derived asthenocormines including *Ohmdenia* and the edentulous taxa. Fringing fulcra are shown here to also be absent in the caudal fin of *Martillichthys* (Supplementary File S1).

Procurrent fin rays account for all caudal lepidotrichia placed anterior to the longest fin ray (first principal ray) of the leading edge [14] and are immediately succeeded by the principal fin rays. In *Germanostomus pectopteri* gen et sp nov., they number approximately 14 in the hypaxial lobe (compared to at least 16 in *S. esocinus*), where they articulate with the first preural haemal spine and the anteroventral region of the hypural plate. The number of dorsal procurrent rays is unknown due to the incompleteness of the hypural region and the moderate disarticulation of the distal portion of the epaxial lobe, which has obscured the boundaries between the procurrent and principal caudal rays. Procurrent rays are segmented medial-distally with the terminal procurrent rays along the hypaxial leading edge showing the 'pattern A' fringing fulcra. Similarly, as in *S. esocinus*, the terminal segments of the epaxial procurrent rays form lancet-like terminations, as opposed to bearing fringing fulcra along the leading edge.

Principal fin rays include the caudal lepidotrichia placed posterior to the longest fin ray, and number approximately 18 in the lower lobe and at least 15 in the upper lobe. All principal rays comprise stiff, elongated lepidotrichia, which asymmetrically bifurcate free from joints close to the distal margin. The principal rays articulate and radiate outwards from the hypural plate, towards the distal-most extremities of the leading edges in each lobe, as well as forming the entirety of the posterior margin of the caudal fin. The terminal four principal rays, which protrude horizontally posterior to the hypural plate and lay within the buttress of the upper and lower lobes, present a different morphology. These rays are much shorter and are each medially bifurcated several times to form wide and delicate posterior fans, as in *Saurostomus esocinus* and *Asthenocormus* [24].

The basal dorsal scute, which serves as an anterior buttress for the epaxial basal fulcra for increased caudal rigidity, is small and ovoid with a width equating to roughly two-thirds of its length. The bone is dense and subtly convex with a prominent median dorsal ridge (Figure 7D,G) and is ornamented at its anterior margin by several deep longitudinal grooves similar to those present in the gular, pelvic plates, and dorsal scute of *S. esocinus*. The anterior notch, which creates a bifurcating split between the left and right anterior portions of the basal dorsal scute in *S. esocinus,* is not present in *Germanostomus pectopteri* gen. et sp. nov. Interestingly, immature specimens of *S. esocinus* display a much narrower anterior notch, wherein the bifurcation only becomes more prominent with increasing body size, being well developed in mature individuals (≥1000 mm [24]; Figure 7F). By comparison, the basal dorsal scute of *Ohmdenia multidentata* (Figure 7H and Figure S9D) is quadrilateral with an inverted isosceles trapezoid-like shape, but it is also characterized by a prominent median dorsal ridge and absence of a bifurcating anterior notch. However, in contrast to *Germanostomus pectopteri* gen. et sp. nov., the dorsal margin of the scute is relatively wider in *O. multidentata* and is characterized by an anterior extension of the median dorsal ridge, creating an anteriorly projecting medial spine (Figure 7G). This projecting spine is highly unusual and is not present in any other examined pachycormid taxa. The dorsal scute in *Martillichthys renwickae* [13] (misinterpreted as the first preural vertebra in [13]: figure 2C) is twice as wide as long, with an expanded pair of lateral processes and a lack of anterior ornamentation, but it does retains a shallow, although slightly elevated anterior notch (Figure 7I). A similar dorsal scute morphology is seen in *Asthenocormus titanius* [1,76] (Figure 7J). A description of the caudal fin of *Martillichthys*, including the basal dorsal scute, is provided in Supplementary File S1 (Figure S2).

*Pre-caudal scaly keel and squamation.* A pre-caudal scaly keel (pcsk) comprising a series of highly specialized and enlarged lateral scales forming a lateral fan-like projection on the caudal peduncle, similar to that of *Saurostomus esocinus* (see [24], figure 16), is present in the new species (Figure 7A–C). Both sides of the keel are slightly displaced from their life position on the caudal peduncle, with the left keel scales partially overlying the epaxial lobe and the right keel scales displaced downwards in front of the fringing fulcra of the hypaxial leading edge. The right side is best preserved and is composed of seven large (ca. 10 mm) rhombic- to diamond-shaped scales that are longitudinally aligned. On the left side, up to ten scales are present, and although they have been more aggressively prepared, nonetheless they retain their original articulation. The pre-caudal scaly keel is a unique structure presently shared only with *Saurostomus esocinus*, and is morphologically unique from the non-homologous (see Reference [24]) scaly caudal apparatus scales present in a few Middle and Late Jurassic pachycormids (*Orthocormus roeperi*; *O. cornutus*; *Hypsocormus* spp., and *Sauropsis longimanus* [6,14,15] as well as from the dissimilar ring of small scales present on the caudal fins of *Pachycormus* [6,14,63] and *Orthocormus? tenuirostris* (NHMUK PV P 10906; pers obs. SC).

Squamation is mostly absent in the both the holotype (SMNS 15815) and paratype (SMNS 56344), implying that *Germanostomus pectopteri* gen. et sp. nov., was naked—a condition shared with (at minimum) *Ohmdenia*, *Martillichthys* and *Asthenocormus* ([1,13,16,31]: SC pers. obs.) but not with *Saurostomus esocinus*, which retains reduced and weakly mineralized squamation across the entirety of its body. A single patch of skin on the caudal peduncle of the holotype of *Germanostomus pectopteri* gen. et sp. nov., preserves faint impressions of possible minute (2 mm) rhombic scales, suggesting that some small patches of very delicate squamation may have been present. Damage or loss of the body scales due to decay and/or preparation is unlikely, given the exceptional preservation quality in the abdominal region (see below).

*Gastrointestinal tract and gut contents.* The gastrointestinal tract is remarkably well preserved in the holotype, revealing details of the soft tissue anatomy including the alimentary canal as well as gut contents associated with the fish's final meal (Figure 8A). The periphery of the gut is preserved as a dark organic film, thus providing an indication for its size and placement within the abdominal cavity—originating just behind the pectoral fins and extending posteriorly, occupying most of the abdominal region and terminating a few centimeters before the origin of the anal fin. The gut measures approximately 200 mm in length and has a superficially elliptical profile, although it is uncertain if this shape has been manipulated by internal decay processes. Anterior of the gut there is a wide space between the pectoral fins where the esophagus should theoretically be situated. This region is better preserved in the paratype (SMNS 56344), in which the foregut cavity is infilled by an organic mass heavily enriched with hooklets of an indeterminate coleoid cephalopod (Figure 8B).

The alimentary canal is mostly complete and is well preserved in epirelief as a phosphatized cololitic infill. Regionalization of the midgut (main intestine) and hindgut (spiral valve and anal cavity) portions of the intestine are easily differentiated, with a combined intestinal length of 190 mm in the holotype. As a whole, the intestine is straight (as opposed to folded in derived teleosts [86–88]) with a slightly undulated trajectory along the abdominal cavity, extending well behind the posterior periphery of the gut and terminating close to the anal region (Figure 8A). The midgut portion of the intestine occupies the medioanterior region of the abdomen and is composed of a single widened canal, which is strongly sigmoidal longitudinally. The hindgut is imperfectly complete, with only the spiral valve preserved by cololite infill. The spiral valve is a localized structure in the hindgut formed by infolding of the intestinal mucosa in a longitudinal spiral pattern to increase surface area to optimize nutrient absorption [86,88,89]. In *Germanostomus pectopteri* gen. et sp. nov., the spiral valve is tightly folded, with a minimum of sixteen rotations. The structure is confluent with and fractionally longer than the midgut intestine but extends well behind the posterior periphery of the gut region (indicated by the black organic film),

becoming narrower and deflecting gradually downwards towards the anus. The narrow void between the distal tip of the final rotation of the spiral valve and the anal fin marks the location of the rectal cavity, although this final portion of the canal is not preserved due to an absence of cololitic material in this region.

Both the gut and intestinal regions of the alimentary canal preserve non-identifiable remains of both coleoids and small actinopterygians, demonstrating a lack of specificity in the diet of *Germanostomus pectopteri* gen. et sp. nov. Small chitinous hooklets and phosphatic fragments of indeterminate belemnoid (diplobeliid) coleoids are present throughout the alimentary canal, although these are more greatly concentrated in the foregut region. Coleoid remains are rarer in the foregut of the holotype than that of the paratype, with surviving hooklets showing evidence of chemical corrosion, suggesting that the squid meal was mostly digested at the time of the fish's death. By contrast, hooklets preserved within the foregut of the paratype are unaltered and are highly abundant, forming dense organic masses and suggesting that the coleoids were ingested shorty before the carcass became buried. Isolated fish scales measuring between 0.5 and 2 mm are also preserved within the gut and mixed within the phosphatized intestine. These larger scales possess a thin ganoin layer and are strictly rectangular with a single serrated margin. When compared to all known actinopterygians in the Posidonienschiefer Formation, they best match the abdominal scales of †*Pholidophorus* sp.

## 4. Discussion

### 4.1. Phylogenetic Interrelations of Germanostomus gen. nov., within †Pachycormiformes

*Germanostomus pectopteri* gen. et sp. nov., is confidently identified as a pachycormid due to the combination of the following synapomorphies (based on Lambers [1]): presence of a rostrodermethmoid forming anterior border of upper jaw; left and right nasals separated by rostrodermethmoid; premaxilla immobile; supramaxilla reduced and placed dorsoposterior to the partially mobile maxilla; powerful lower jaw articulating well behind the orbit; low coronoid process on the mandible; single unpaired gular; branchiostegal rays plentiful; stiff unsegmented pectoral fins; fins rays distally bifurcate asymmetrically and free from joints; vertebral column weekly ossified; supraorbitals absent; and hypurals fused into a single hypural plate. This referral is supported by our second analysis (Gouiric–Cavalli and Arratia [6] matrix) in which *Germanostomus* is resolved within Pachycormidae rather than in one of the other included actinopterygian lineages.

The first analysis, based on a modified version of the Friedman [16] matrix, produced a strict consensus of 12 trees. Resolution of the cladogram firmly places *Germanostomus pectopteri* gen. et sp. nov., between the more basal *Saurostomus esocinus* and *Ohmdenia multidentata*, occupying an intermediate phylogenetic position between these early-diverging asthenocormines (Figure 9). *Germanostomus pectopteri* gen. et sp. nov., therefore clusters more closely than *Saurostomus esocinus* to the suspension-feeding clade [24]. The resolution of *G. pectopteri* sp. nov., as between *Saurostomus* and *Ohmdenia*, rather than a sister species to *Saurostomus esocinus*, combined with a suite of diagnostic characteristics (see Diagnosis) supports our assignment of this new species to the new genus *Germanostomus* rather than as a second species of *Saurostomus*.

The inclusion of *Germanostomus* did not change the overall pachycormid cladogram substantially, although resolution within the suspension-feeding clade was further reduced (Figure 9). The resolution of this clade was already poor in previous analyses, with *Rhinconichthys* spp., *Martillichthys* and *Bonnerichthys* in a polytomy sister to a smaller clade containing *Leedsichthys* and *Asthenocormus* [24]. However, this latter clade has now also collapsed, with all suspension-feeding taxa reduced to a single polytomy. The cause of this collapse is uncertain but was likely influenced by our partial re-scoring of *Martillichthys* and *Asthenocormus* based on newer literature [20] and observations.

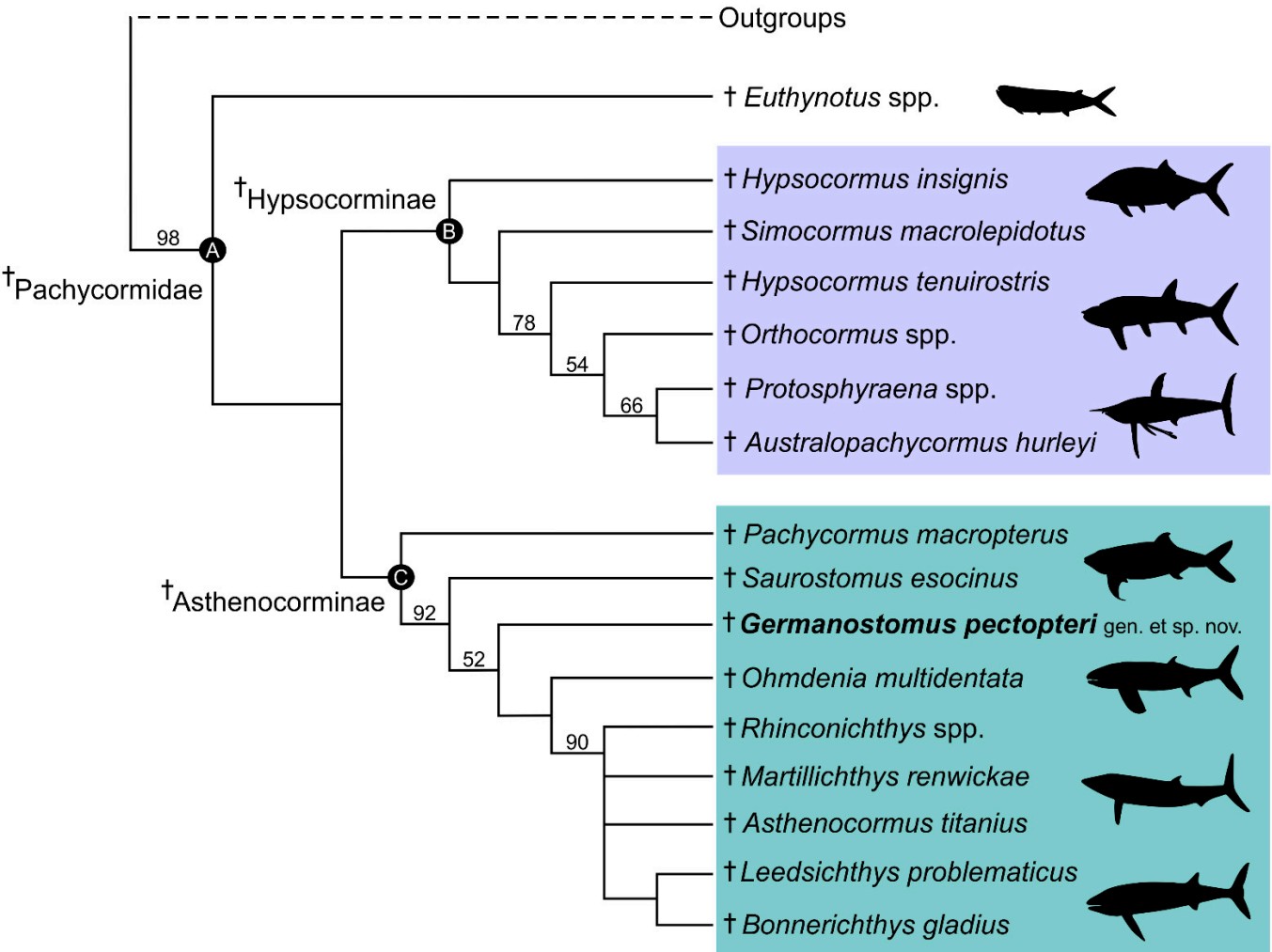

**Figure 9.** Strict consensus of 12 most parsimonious trees (MPTs) composed of 30 taxa and 129 characters based on the matrix of Friedman [16], indicating the position of *Germanostomus* gen. et sp. nov. within Asthenocorminae. Jackknife support values greater than 50% are indicated. Colour boxes indicate grouping of the two sub-family clades, Hypsocorminae (node 2) and Asthenocorminae (node 3). Pachycormidae (node 1) includes both Hypsocorminae and Asthenocorminae in addition to the basal genus *Euthynotus*. Non-pachycormid outgroups have been pruned from the figure (see Supplementary data for full list and expanded cladogram). All silhouettes except for *Leedsichthys* (redrawn from Reference [90]) were produced by the authors.

Our second analysis, in which *Germanostomus pectopteri* gen. et sp. nov. was scored in the matrix of Gouiric–Cavalli and Arratia [6], produced a slightly different result. *Germanostomus* was resolved as the sister of *Saurostomus esocinus*, together forming a clade at the base of Hypsocorminae. As with the analysis of the modified Friedman [16] matrix, inclusion of *Germanostomus* did not change the overall pattern of relationships within Pachycormidae. Gouiric–Cavalli and Arratia [6] included far more characters and nonpachycormid actinopterygians than Friedman [16] but omitted key taxa in the discussion of the evolution of suspension feeding, in particular *Ohmdenia*, since they were most interested in resolving the relationships of a hypsocormine (*Kaykay*). *Ohmdenia multidentata* has previously been established as sister to the suspension-feeding clade [16,17,20,24], and its presence in the matrix is expected to influence the position of *Saurostomus*. In addition, the Gouiric–Cavalli and Arratia matrix omits a few of the key characters associated with the evolution of suspension-feeding capabilities, notably the presence or absence of the opercular process

on the hyomandibula; these are included in Friedman [16]. Our preferred tree is therefore that derived from the Friedman [6] matrix.

### 4.2. Affinities with the Suspension-Feeding Clade

The suspension-feeding clade is a monophyletic group within Asthenocorminae subfam. nov. that contains all edentulous pachycormids, including *Asthenocormus*, *Leedsichthys*, *Martillichthys*, *Rhinconichthys*, and *Bonnerichthys*. Although true suspension-feeding did not evolve until the Middle Jurassic (Callovian), as indicated by a shift towards skull elongation and edentulousness [5,13,18,20,90], Asthenocorminae diverged from Hypsocorminae prior to the Early Toarcian stage of the Early Jurassic [16,24]. Early-diverging asthenocormines (*Pachycormus*, *Saurostomus*, *Ohmdenia*, *Germanostomus* gen. nov.) are not edentulous. Although *Pachycormus* is recovered as the most basal asthenocormine in our preferred analysis, this result is very poorly supported. *Pachycormus macropterus* does not share synapomorphies of the trophic apparatus with the suspension-feeding clade, but the absence of the pelvic fins is a synapomorphy shared by all asthenocormines, including *Pachycormus macropterus* [1,27,33]. The position of *Saurostomus esocinus* is much better supported, including the following synapomorphies: expanded anterior corpus on the parasphenoid (after Reference [20]); absence of the suborbital and infraorbital bones (not recorded or irrefutably absent in all successive taxa); large body size ($\geq$1500 mm) with an elongated post-cranial portion; highly reduced ossification in the vertebral column (neural and haemal arches only); differentiated dorsal basal scute that is short and wide (as opposed to elongate and narrow in *Pachycormus* and Hypsocorminae) [24]. All of these synapomorphies are also shared with the new taxon *Germanostomus pectopteri* gen. et sp. nov., with the exception to the anterior portion of the parasphenoid which is not preserved, thus strongly supporting its phylogenetic position as an early-diverging member of Asthenocorminae.

*Germanostomus* also displays synapomorphies with more derived asthenocormines to the exclusion of *Saurostomus*. These include:

1. Hyomandibula that is tall and superficially rectangular with a posterior lamina. The hyomandibula of *Saurostomus* is strongly waisted and does not possess a posterior lamina; in *Ohmdenia* the bone is perfectly rectangular with a well-developed posterior lamina.
2. The opercular process of the hyomandibula is entirely absent in *Germanostomus*, *Ohmdenia*, and most successive taxa with the possible exception of *Bonnerichthys* and *Martillichthys* (see Section 4.3.4); this process is well developed in *Saurostomus*.
3. Scales highly reduced and possibly absent over much of the body in *Germanostomus*.
4. Increased elongation of the upper and lower jaw relative to the posterior skull (see Section 4.3.2). This character is more extreme in *Ohmdenia*, but in *Germanostomus* there is already significant elongation of the upper and lower jaw in comparison to *Saurostomus esocinus*, including disproportionate elongation of the premaxilla (relative to maxilla) and the dentary (relative to the angular and surangular).

Elongation of the jaws and gill basket is a necessary mechanical precursor of suspension-feeding, as it allows gape increase and compensates for the reduction and eventual loss of teeth in the jaws [5,13,16,17,20]. *Ohmdenia multidentata* is remarkable for possessing both hypertrophically elongated jaws and retaining well-developed *Saurostomus*-type dentition on the dentary and the anteriorly inflated coronoids (Figure S9C), and is a critical taxon uniting *Saurostomus* and *Germanostomus* with the suspension-feeding clade. The relative elongation of the anterior upper and lower jaws (premaxilla relative to the maxilla; dentary relative to the angular and surangular) in *Germanostomus* compared to *Saurostomus* suggests a gradual transition towards the degree of jaw elongation seen in *Ohmdenia*.

Fringing fulcra are absent in the caudal fins of *Asthenocormus titanius* [1,63], *Martillichthys* (see Supplementary File S1 for a description of the caudal fin and Figure S2), and likely also in *Leedsichthys problematicus* [18,21,91], suggesting that their absence might be a hitherto undetected synapomorphy of the suspension-feeding clade. However, the

caudal fin is either unknown or poorly preserved in most edentulous taxa. *Saurostomus esocinus* and *G. pectopteri* both possess hypaxial fringing fulcra, although these fulcra are comparatively larger in *S. esocinus,* where they are often hypertrophied, but are contrastingly reduced and smaller in size on the hypaxial lobe of *G. pectopteri* gen. et sp. nov. (Figure 7B,C). Caudal fringing fulcra are not observed in *Ohmdenia multidentata,* although this may be attributed to incompleteness in the caudal region in the holotype (Figure S9A).

In addition to the synapomorphies discussed above, *Germanostomus* shares several key characteristics with *Ohmdenia multidentata,* which are absent in *Saurostomus esocinus*:

(1) The pectoral fin shape is broad with a characteristically wide and obtusely rounded distal margin. This morphology is also present in *Leedsichthys* [12,13], and it is strongly differentiated from the conventional falciform shape with a sharply pointed distal tip in the pectoral fins of *Pachycormus* and *Saurostomus.*

(2) The differentiated dorsal scute is shield-like with a prominent median dorsal ridge, and the anterior notch of the dorsal basal scute is absent. By contrast, the dorsal scute of *Saurostomus* does possess an anterior notch that serves to bifurcate the anterior portion of the scute, but there is no median ridge on the external surface. In *Martillichthys* and *Asthenocormus,* the dorsal scute is extremely short but laterally expanded, with the width of the dorsal scute accounting for twice the length. The lateral margins are hypertrophically elongated to form 'wings' (Figure 7I,J) while at the medial-anterior margin these is a narrow but a well-developed concave anterior notch. The dorsal surface is marked by a 'Y'-shaped median ridge in *Martillichthys* (Figures 7I and S2), but this is apparently absent in *Asthenocormus*. The dorsal basal scute is unknown for *Leedsichthys*, *Rhinconichthys* spp., and *Bonnerichthys*. The dorsal scute of *Pachycormus* is long and narrow, with a superficial diamond-shaped outline that is equal in size to, and is thus often difficult to differentiate from, the epaxial basal fulcra [63].

Therefore, we conclude that *Germanostomus pectopteri* gen. et sp. nov. shares all of the synapomorphies for the suspension-feeding clade present in *Saurostomus*, in addition to several additional characters previously recorded in *Ohmdenia* (e.g., Reference [16]).

## *4.3. Comparison of Germanostomus to Other Pachycormid Fishes*
### 4.3.1. Dentition

Traits related to the dentition (e.g., tooth size, tooth shape, tooth ornamentation and number of tooth rows) have important taxonomic value [6,92,93], with their morphological diversity providing a more comprehensive view of actinopterygian synecology [6,94–96]. The morphological diversity of pachycormid dentitions is poorly understood [6], as these have only been partially described in the literature for many taxa [1,23,27,30,33]. Dental morphology is highly disparate across Pachycormiformes, ranging from small isodont (e.g., *Euthynotus*, *Pachycormus*, [26,27,33]), villiform macrodont (*Saurostomus*, *Ohmdenia* [16,24]), to front-fanged macrodont (e.g., *Hypsocormus*, *Orthocormus*, *Simocormus*, *Kaykay* [1,6,14,15]); however, the functional morphology and evolution of pachycormid dentitions are poorly studied in the literature.

The teeth of *Germanostomus pectopteri* gen. et sp. nov. are morphologically isodont, comprised of simple, unornamented conical crowns with slightly widened bases, uncurved apexes and fine, pointed acrodin caps, which are all equal in size and are evenly spaced along the entirety of the upper and lower jaws. A smaller marginal tooth row comprising minute (less than 40% of principal row teeth) isodont crowns extends along the external periphery of the lower jaw. Teeth on the palate and branchial plates (if indeed present) are unknown in *Germanostomus pectopteri* gen. et sp. nov. Following the revised classification of piscivorous actinopterygian dentitions of Mihalitsis and Bellwood [95], the dentition of *G. pectopteri* gen. et sp. nov. is classified as villiform. Upon occlusion, teeth in the upper and lower jaws would have exerted a mode-rate stress upon the prey and functioned predominantly in prey capture, but without the large front-fanged macrodont teeth seen in some other pachycormids (e.g., *Orthocormus*, *Simocormus*) [1,15], *Germanostomus* was likely incapable of prey processing [95]. Piscivorous actinopterygians with low prey

processing capabilities are thus reliant on swallowing their prey whole [94–96]. Due to their relatively large size, the teeth of *Germanostomus* would have been ideal for fast capture and piercing of elusive prey with irregular or penetration-resistant integument, such as coleoid cephalopods, crustaceans, and smaller fishes [95,96], including actinopterygians with small ganoid scales such as *Pholidophorus*.

The teeth of the closely related *Saurostomus esocinus* are relatively similar to those of *Germanostomus*, but are far more numerous in the principal tooth rows, with each tooth bearing diagnostic ornamentation at the crown bases consisting of deep longitudinal folds [24]. Tooth crowns are always strongly recurved in sub-mature (75 cm–99 cm) and mature (≥1 m) specimens of *Saurostomus esocinus* [24]; a feature not seen in either *Germanostomus* (Figures 3 and 4) or *Ohmdenia* [16] (Figure S9B,C). Immature specimens of *Saurostomus esocinus* (≤74 cm), including the missing holotype ([24,29]: Figure 1A,B), possess fewer teeth, an unerupted coronoid tooth plate, and display apicobasally straight tooth crowns, similar to those present in the new taxon. Juvenile specimens of *S. esocinus* do however strongly display longitudinal folds around the tooth base as described above. The larger body size (>1 m), unornamented tooth crowns, and greater mandible length to depth ratio confidently differentiates the holotype of *Germanostomus pectopteri* gen. et sp. nov. from all ontogenetic stages of *Saurostomus esocinus*.

*Ohmdenia multidentata* is a much larger fish, with a mandible length to depth ratio more than double that of *Germanostomus pectopteri* gen. et sp. nov. The numerous overcrowded teeth in the mandible of *Ohmdenia* are significantly thicker and stouter, with fine acrodin caps which are generally blunter rather than pointed (Figure S9B). Interestingly, teeth on the principal mandible tooth row show a morphological gradient [31], with the mesial teeth being narrower, slightly recurved and more pointed, whilst the teeth become progressively stouter and blunt distally, as described above. This evidently functional heterodonty may be associated with the elongation of the skull and a change in diet or feeding style, potentially resulting in an exaptation for suspension-feeding [16,20,24]. In addition, the mesial teeth in both the mandible and possibly the maxilla/?premaxilla of *Ohmdenia* display subtle ornamentation of the tooth bases, similar although not identical to the folding seen in *Saurostomus esocinus*.

Both *Saurostomus* and *Ohmdenia* share an unusual feature in their lower jaws whereby the anterior most villiform teeth are projecting forward up to 180° to the anterior-posterior axis of the jaw. This characteristic is shared with members of the toothed clade (notably *Protosphyraena* and *Orthocormus*) but absent in more basal pachycormids (*Euthynotus*, *Pachycormus*, *Sauropsis* spp.) as well as in the new taxon *Germanostomus pectopteri* gen. et sp. nov.

4.3.2. Elongation of the Premaxilla

The premaxilla is comparatively longer relative to the maxilla in *Germanostomus* than in *Saurostomus*, occupying approximately 33 % of the total upper jaw length in *G. pectopteri* compared to only 17% in *Saurostomus esocinus*. The immobile premaxillae of *Saurostomus* and more basal taxa such as *Pachycormus* [23,26,27] and *Euthynotus* [33] are very short in relation to their upper jaw lengths, and support very few teeth along their dentigerous margins. No more than four teeth are present on the premaxilla (excluding lateral marginal teeth) of *Saurostomus esocinus*, best observed in SMNS 12576 (see figure 5D in Reference [24]). For *Germanostomus pectopteri* gen. et sp. nov., the comparatively longer premaxilla holds a higher number of teeth, with no less than eight present in the holotype (Figure 4)—approximately double that of *S. esocinus*. In *Germanostomus* gen. nov., the premaxilla has expanded lengthwise to occupy the entire length of the antorbital and terminates in line with the posterior margin of the nasal, immediately before the orbit (Figure 4). The premaxilla is much shorter and is placed predominantly anterior of the antorbital in *Saurostomus*, in which the premaxilla only slightly extends under the nasal-antorbital contact (Reference [24]: figures 4 and 9B).

The elongation of the premaxilla is not only an autapomorphy of *Germanostomus* gen. nov., but provides vital insight into the evolution of the skull in derived asthenocormines.

Elongation of the skull through lengthening of both the upper and lower jaws is required for development of suspension-feeding capabilities, as indicated by modern analogues (e.g., whale sharks, basking sharks, and mysticete whales) [5,13,17,18,20]. The premaxilla is poorly known in almost all of the suspension-feeding pachycormids, mostly due to their poor ossification (e.g., [5]). The premaxilla in *Ohmdenia* is unknown; however, it is not necessarily missing in the holotype, as there are numerous fragmented toothed elements in the skull region which are problematic to identify [16]. A small, elongated bone in the antorbital region of the *Martillichthys renwickae* holotype (NHMUK PV P 61563) was tentatively identified as a partial premaxilla [20]. A premaxilla is not reported for any of the Late Jurassic or Cretaceous suspension-feeding pachycormids, despite a handful of mostly complete and well-articulated skulls (e.g., *Rhinconichthys* spp. [17]), and the premaxilla is demonstrably absent in *Asthenocormus titanius* [1]. The relative space occupied by the premaxilla can be approximated by the length of the maxilla relative to the lower jaw. This reveals a complex pattern in the evolution of the upper jaw in Asthenocorminae. In *Saurostomus*, the maxilla is approximately 70% of the length of the lower jaw; however, in *Germanostomus* this drops to 54%, due to the longer premaxilla. A proportionately short maxilla is also observed in *Martillichthys* and *Asthenocormus*, from the Middle and Late Jurassic, respectively [1,20]. The anteriorly short, potentially free maxilla may have contributed to increase gape size, as interpreted for extant polydontids [97]. In the Cretaceous taxon *Rhinconichthys*, the maxilla is even more proportionately elongate than in *Saurostomus* [17], suggesting a complex evolutionary trajectory for upper jaw morphology. We find the following scenario to be most probable: during the initial lengthening of the skull in basal asthenocormines, the length of the premaxilla increased relative to the maxilla. In the Late Jurassic suspension feeders, ossified premaxillae were reduced or lost entirely, along with other regions of the skull (e.g., postorbitals, suborbitals, and dermosphenotic) as part of the further reduction in cranial ossification present in these edentulous taxa [1,5,13,18,20,24]; however, the maxillae remained short. Given its phylogenetic position, *Rhinconichthys* can either be interpreted as retaining the basal state (as in *Saurostomus*) or having undergone a secondary elongation of the maxilla. Given its specialized hyomandibular morphology, we find the latter scenario to be more probable.

### 4.3.3. Pectoral Fin Morphology

The large pectoral fins are perhaps the most salient feature of *Germanostomus pectopteri* gen. et sp. nov. to differentiate it from all other described members of Pachycormiformes. The inverted 'D'-shaped morphology formed from a distal-posteriorly curved anterior rod and all lepidotrichia terminating along a perfectly even transverse plane, combined with a minute lobe-like posterior fillet, confidently distinguishes the pectoral fin shape from all other pachycormid fishes. The three most anterior fin rays are fused into a stiffened, and obtusely curved anterior rod which extends the entirety of both the (anterior) leading edge and encapsulates the entire distal (ventral) margin of the fin. Fusion of the anterior pectoral fins rays along the leading edge is considered a derived trait in †Pachycormiformes [98,99] and is shared among all asthenocormines including *Pachycormus* [5,16,24,99]. This trait also appears to have arisen convergently in some hypsocormines, notably in the pectoral fins of *Orthocormus* spp. [1,14] and *Protosphyraena* [58,60]. The anterior rod in *Saurostomus esocinus* is composed of at least the first three pectoral lepidotrichia, is relatively straight along much of the anterior leading edge and extends posterodistally to form a prominent 'whip-like' trailing edge for improved hydrodynamic velocity and drag reduction [24,65]. The anterior rod is obtusely curved to encapsulate the entirety of the distal fin margin in *Germanostomus*, with the distal termination of the anterior rod situated along the same plane as the other pectoral lepidotrichia (Figure 6C). The anterior rod is also well developed in *Ohmdenia multidentata*, where it is composed of a fusion of at least the first six pectoral lepidotrichia (Figure S9E). Similar to *Saurostomus*, the anterior margin of the fused rod is straight and does not share the strong distal posterior curvature as in *Germanostomus* gen. nov. The distal margin of the pectoral fin in *Ohmdenia* is also rounded (albeit not

as greatly), with posteriorly sweeping lepidotrichia similar to *Germanostomus*; however, the anterior rod does not curve posteriorly to encapsulate the distal margin, rather the anterior rod of *Ohmdenia* terminates sharply at the distal-anterior margin (Figure S9E). The pectoral fin of *Martillichthys renwickae* from the Middle Jurassic of England is of a similar morphology to that of *Saurostomus esocinus*; notably its strongly falciform shape with a markedly concave trailing edge topography and straight and elongated anterior rod which does not cover the distal fin margin [13,65] (pers obs. SC). The anterior rod is also anteriorly straight and is not curved posteriorly, similar to all other edentulous suspension-feeders (*Bonnerichthys*, *Rhinconichthys*, *Asthenocormus*, *Leedsichthys*: [1,5,13,17,18,99]). However, similar to *Germanostomus*, an almost straight posterior margin is seen in some of the more derived edentulous pachycormids, notably *Bonnerichthys* [5,99] and *Asthenocormus* [1,13]. The presence of a strong posteriorly curved anterior rod encapsulating the entirety of the distal margin of the pectoral fin is therefore an autapomorphy of *Germanostomus pectopteri* gen. et sp. nov.

*Pachycormus macropterus* also possesses a similarly thickened anterior fin ray, al-though it is not always formed by fusion of multiple rays. This enlarged anterior ray of *Pachycormus* does not extend along the entirety of the leading edge, rather it is succeeded distally by minute fringing fulcra ('special fulcra' in Reference [6]) in well-preserved specimens ([33]: figures 60 and 61). However, as in *Germanostomus*, the posterior fin margin is also straight in some examples of *Pachycormus macropterus* (e.g., [33]: figure 60). Pectoral fringing fulcra are also observed in the Lower Jurassic pachycormids *Euthynotus* spp. [33] and *Sauropsis veruinalis* (SMNS 87736: pers obs. SC). Pectoral fringing fulcra are absent in *Saurostomus*, *Germanostomus* gen. nov., *Ohmdenia*, and all suspension-feeding taxa; however, the leading pectoral fin edge of *Bonnerichthys* shows an autapomorphic arrangement whereby the fused lepidotrichia are thickened, with the leading edge formed by 'a sharp keel with irregular excavations' [99].

The posterior fillet is characterized by a narrow strip of distal lepidotrichia which forms a posterior projection or expansion of the proximal-posterior margin of the pectoral fin [64]. Not all pachycormids share a posterior fillet, although those that do demonstrate a high degree of morphological disparity between species. For instance, *Saurostomus* shows two separate posterior fillet morphologies, interpreted as being related to intraspecific variation or specimen taphonomy: (1) a well differentiated, projecting blade-like morphology (morphotype 1); and (2) a more weekly differentiated, deeper, and convex morphology (morphotype 2) [24]. Morphologies of the posterior fillet, if indeed originally present, are unknown in the incompletely preserved pectoral fins of *Ohmdenia multidentata* and *Leedsichthys* [5,64] (pers obs. SC), with this structure almost certainly absent in *Asthenocormus* [1]. A posterior fillet is present in the Callovian pachycormid *Martillichthys renwickae*; however, the constituent fin rays are slightly displaced from their life position in the holotype (NHMUK PV P.61563), which has led to their misidentification as pelvic fins [12,13,100]. We did not observe any evidence for pelvic fins in this taxon.

According to Reference [65], the pectoral fin of *Sauropsis* spp. has a gladiform morphotype, with a posterior fillet allegedly absent for this genus. However, a small posterior fillet has been reported for †*Sauropsis longimanus* [6], as well as the Early Jurassic species †*Sauropsis*? *veruinalis* [55], which also displays a differentiated, albeit smaller posterior fillet (NHMUK PV P 13006; pers obs. SC). An alleged small posterior fillet was also reported for †*S. depressus* in Reference [14]; however, the specimen their observation was based on (JME SOS 2181) is a misidentified *Orthocormus* sp. [15]. Most recently, a small posterior fillet was noted in the Tithonian hypsocormine †*Kaykay lafken* [6]. A partial posterior fillet is also present on the Munich specimen of †*Pseudoasthenocormus retrodorsalis* (BSPG 1956 I 361) (S.C. pers. observ.).

### 4.3.4. Morphological Disparity of the Hyomandibulae

The hyomandibula is a well ossified and morphologically disparate bone in the skull of pachycormids and has both functional [101] and likely taxonomic value [6,102]. All

pachycormids possess a dorsoventrally elongated hyomandibula with a differentiated dorsal and ventral head for powerful articulation between the hyoid arch and the skull roof. When considered across the clade, certain hyomandibular morphologies are associated with particular inferred feeding niches. Basal pachycormids, such as the Toarcian taxa *Euthynotus* spp. and *Sauropsis*? *veruinalis*, as well as most members of the macrophagous clade (Hypsocorminae), possess a relatively short hyomandibula which is strongly waisted towards the midline and bears a prominent triangular medial-posteriorly projecting opercular process [33,55] (S.C pers. obs.). The hyomandibula of *Pachycormus* is strongly asymmetrical, with a dorsal margin wider than the ventral margin, a very shallow medial-anterior lamina and a rectangular opercular process which is set closer to the dorsal margin [20,22,27]. In *Saurostomus esocinus*, the hyomandibula is symmetrical and slightly more elongated dorsoventrally than in *Pachycormus*, with a well-developed medial-anterior lamina and hatchet-like opercular process set just below the expanded dorsal margin. The dorsal and ventral margins are equally wide, and strongly convex [24]. By contrast, the hyomandibula of *Ohmdenia* is strongly rectangular with a greater height-to-width ratio than *Saurostomus* [16] (Figures 5D and S9F). The opercular process is absent in *Ohmdenia*; however, it retains both a wide medial-anterior lamina, and a very narrow marginal medial-posterior lamina as shared with *Germanostomus*. The hyomandibula of *Germanostomus pectopteri* gen. et sp. nov. is medially waisted with equally expanded dorsal and ventral heads, similar to *Saurostomus*, although differing from *Saurostomus* by the presence of a well-developed medial-posterior lamina, which is almost as large as its anterior counterpart, and the absence of an opercular process. Furthermore, the topography of the dorsal and ventral heads in *Germanostomus* are narrower and taller, similar to their shape in *Ohmdenia*. Therefore, the narrower distal heads, combined with a shared absence of the opercular process and presence of both an anterior and posterior medial lamina, more closely unite the hyomandibula of *Germanostomus* to *Ohmdenia* than with *Saurostomus*, with *Germanostomus* seemingly representing a morphological intermediate between the two.

The edentulous clade of pachycormids displays a high level of morphological variation in the hyomandibulae [20]; however, none are closely compatible with *Germanostomus*. The hyomandibula of *Leedsichthys problematicus* (NHMUK PV P 11823) is massive and thick, and displays a prominent dorsal head that laterally expands dorsolaterally from the midpoint, with an undulated dorsal margin and a narrower, more semicircular ventral head. A narrow and short medial-anterior lamina is retained, although the posterior lamina and an opercular process are absent [18,20] (Figure 5). The hyomandibula in *Martillichthys renwickae* is thin and plate-like, with a distinct irregular asymmetry. The ventral portion of the bone appears to be significantly wider than the dorsal portion; an opercular process was interpreted as having originally been present in Reference [20]. This interpretation was based on the subtle thickening of the bone along the broken dorsoposterior margin of the hyomandibula [20]. This should be approached with caution, though, as the posterior bone margins are also thickened in the hyomandibulae of some taxa, which do not possess a process (*Ohmdenia* and *Leedsichthys*). We therefore score the presence of an opercular process as unknown in *Martillichthys*. A medial-posterior lamina appears to have been absent in *Martillichthys*, but a small thinner region at the dorsal-anterior margin may possibly represent a residual medial-anterior lamina (Figure 5G). The shape of the hyomandibula is, by contrast, highly derived in the Cretaceous taxa *Rhinconichthys* spp., where it displays the greatest height-to-width ratio of any pachycormiform [5,17]. The bone is well elongated and 'lever-like', with a narrow dorsal and ventral head, which are strongly rounded and do not possess either medial laminae, medial waisting, or an opercular process (Figure 5H). Reference [17] attributed the function of this derived morphology as a specialized lever for ventrolateral displacement of the jaws during opening to provide optimal expansion of the buccal cavity to aid suspension feeding. Such a derived specialization of the hyomandibula is not seen elsewhere in the clade, with the possible exception of *Bonnerichthys* (Figure 5I). The latter differs from *Rhinconichthys* in the presence of a subtle medial-anterior lamina and a well-developed opercular process of the hyomandibula, similar in size and placement

to *Pachycormus* and *Saurostomus* [5]. The long stratigraphic and phylogenetic gap between *Bonnerichthys* and *Saurostomus*-grade pachycormids from the Lower Jurassic implies that the opercular process may be secondarily acquired in the Late Cretaceous giant. It is noteworthy that both *Saurostomus* and *Bonnerichthys* also share a similar short and wide cranial bauplan [24]—demonstrating further evidence for possible morphological convergence between these two stratigraphically distant pachycormids.

*4.4. Diet and Gastrointestinal Anatomy of Germanostomus Pectopteri gen. et sp. nov.*

The diet of *Germanostomus* was evidently broad and included both fish and cephalopods as inferred from exceptionally preserved gut contents. Chitinous hooklets belonging to an indeterminate belemnoid cephalopod (cf. *Clarkeiteuthis* sp.) form the majority of the gut contents of SMNS 15815 and are especially well preserved in the paratype, where they form the sole component. The remaining gut contents (≈10% total) of SMNS 15815 are composed of isolated actinopterygian scales (0.5–2 mm), with the largest (≤2 mm) attributed to *Pholidophorus* sp.

Coleoids were an abundant prey resource for many predatory actinopterygians in the Posidonia Shale Sea. Non-pachycormid fishes, such as the caturid *Caturus smithwoodwardi* [55] and some marine reptiles (e.g., ichthyosaurs), preserve an abundance of coleoid hooklets in their stomachs [103–105]. A recent review of the diet of *Saurostomus esocinus* ([24]: table 1) revealed that almost all specimens preserving gut contents were either dominated by or consisted solely of isolated belemnoid hooklets, most of which could be attributed tentatively to the diplobeliid *Clarkeiteuthis* sp. A single specimen had the vampyropod *Loligosepia* sp. as gut contents, as well as a minute (2 mm) ammonite larva; accidental ingestion of the latter is probable [24]. Reference [106] provides the first description of coleoid hooklets, attributed to the genus *Phragmoteuthis*, in the gut of *Pachycormus* sp. Several broken belemnite guards were found in association with the abdominal region of the *Ohmdenia multidentata* holotype [31] and were later interpreted as being a chance association [16]. Re-examination of the specimen with use of a hand lens revealed that the guards were partially acid etched, and most were preserved alongside associated hooklets, strongly suggesting that the belemnites indeed represent of gut contents (SC pers. obs.). Teuthophagy was evidently an important dietary niche during the early radiation of Pachycormiformes in the Toarcian, with the inclusion of belemnoid remains in the diet of *Germanostomus pectopteri* gen. et sp. nov. further supporting this assessment.

Piscivory is rarely observed in Lower Jurassic pachycormids. Isolated ganoine scales of *Pholidophorus* sp. are a rare component in the gut contents of *Germanostomus pectopteri* gen. et sp. nov. The majority of *Pholidophorus* specimens from the Posidonienschiefer Formation are between 80 and 220 mm in SL, with their scales measuring between 0.5 and 7 mm, respectively (S.C pers. obs.). Scales preserved in the gut of *Germanostomus* do not exceed a width of 2 mm, suggesting that the prey fish was likely a small individual. The *Pholidophorus* meal was mostly digested and, therefore, was ingested some time prior to the predator's demise. The low abundance of actinopterygian remains in the gut of SMNS 15815 suggests that *Pholidophorus* was a rare or, more likely, an opportunistic prey component in the diet of *G. pectopteri* gen. et sp. nov. The only reported example of a Posidonienschiefer pachycormid with fish remains as gastric contents apart from the *G. pectopteri*-type specimen is a large individual of *Saurostomus esocinus* containing five small *Pachycormus* sp. within the stomach [10,107].

The vertebrate gastrointestinal tract reflects many aspects of organismal biology, including diet, which is crucial to infer trophic occupation and feeding habits, in addition to understanding metabolism, osmoregulation, and nutrient uptake capabilities [88,89,108]. The morphology of the gastrointestinal tract in extinct actinopterygians is hindered by the low preservation potential of soft tissues in the fossil record [88]; rare konservat lagerstätte such as the Posidonienschiefer Formation (in addition to numerous others) offer potentially crucial insight into gastrointestinal anatomy over evolutionary time scales. In osteichthyians, the alimentary canal is divided into three distinct regions: (1) the foregut

comprising the esophagus and stomach, (2) the midgut representing the main intestine following the bile duct, and, lastly, (3) the hindgut or anal region which comprises the rectal cavity and, in some clades, the spiral valve organ [86–89].

The exposed portion of the midgut is well preserved in *Germanostomus pectopteri* gen. et sp. nov., where it is relatively straight (as opposed to coiled or folded), and it is slightly wider than the hindgut portion of the alimentary canal (Figures 2, 5A and 10). The midgut length is comparatively short in relation to that of closely related taxa (see below), with this region of the intestine in *Germanostomus* almost equal in length to that of the hindgut portion. The midgut in *Saurostomus esocinus* (Figure 10) is narrower than the hindgut but is relatively more elongated, with the length of the midgut accounting for 400% of the length of the hindgut. The trajectory of the midgut in *Saurostomus* follows a gentle sigmoidal curvature, with the anterior and medial portions of the intestine placed high in the abdominal cavity. The posterior half of the midgut slopes ventrally to meet the spiral valve, forming a deep convex shape to the alimentary canal (Figure 10). The midgut in both *Saurostomus* and *Asthenocormus* is placed dorsal to the stomach, with the anterior opening of the midgut connected to the dorsal-anterior roof of the stomach [76]. In *Pachycormus macropterus*, the midgut portion of the alimentary canal is placed below the stomach, originating close to the ventral midpoint. The midgut of *Pachycormus* is unusual compared with other pachycormids in that it is tightly folded with sharp acute curves, unlike the conventional straight to convex trajectory in *Saurostomus* and *Germanostomus* (Figures 10, 11 and S10). This morphology is best observed in SMNS 4415 and BRLSI.1838/BRLSI.1384, where the intestine drops ventrally from the stomach and becomes tightly folded in a superficially 'double S' trajectory before straightening out again towards the anal region (Figures 11 and S10C,D). The hindgut, including the spiral valve of *Pachycormus*, is well preserved in SMNS 56230 (Figure S10A,B), where the alimentary canal, infilled with cololite, passes between the left and right preanal scutes and marks the extent of the rectal cavity, which ends immediately before the first basal fulcra of the anal fin. The number of mucosal folds (rotations) in the spiral valve is variable between taxa, with a maximum of 14 in *Pachycormus*, $\geq$16 in *Germanostomus pectopteri* gen. et sp. nov., $\leq$30 in *Saurostomus esocinus* [24]; and $\geq$70 for *Asthenocormus titanius* [109] (Figure 10).

The spiral valve is a regionalized series of tight mucosal folds that run along the intestine in a spiral-like pattern. Despite its name, it is not a true valve, rather it is formed by concentrated infolding of the intestinal mucosa and submucosa, with the purpose of increasing the surface area to optimize nutrient uptake as a compensatory adaptation in fishes with a short alimentary tract [86,89]. A spiral or scroll valve is found in some Paleozoic jawless vertebrates as well as in extant lampreys [88,89,110–113], all extant chondrichthyans, the extinct 'placoderms' [114–117], as well in as most non-teleostean actinopterygians and non-tetrapod sarcopterygians, including coelacanths [115], lungfishes [87], *Polypterus* [89], acipenseriforms [89,118–122], lepisosteiforms [83], the genus *Amia* within amiiforms [68], and several extinct actinopterygians, including saurichthyids [88], pachycormids [24,106], and caturids [68]. Spiral valves are also likely present in pycnodonts [123] (figure 65). Derived Teleostei do not possess a spiral valve; rather their intestines are lined with protruding villi to optimize surface area for increased epithelial absorption of nutrients [89].

The presence of spiral valves in asthenocormine pachycormids is historically well-documented, first reported in Reference [76] for *Asthenocormus* from the Upper Jurassic of Solnhofen. SMNS 15815 (*Germanostomus pectopteri* gen. et sp. nov.) was the first Toarcian pachycormid to be recognized as possessing a spiral valve [52] and has since been figured in numerous works as an example of the presence of this structure in Pachycormiformes (see synonymy list for citations). Among the Early Jurassic taxa, a spiral valve organ has since been reported in *Saurostomus esocinus* [44,106] as well as in *Pachycormus macropterus* [106]. The presence of a spiral valve has not been reported in many hypsocormines, but it is present in *Orthocormus roeperi* [14], *Hypsocormus insignis* (EM + SC, pers. observ.), and likely *Sauropsis* sp. (SC, pers. observ—BSPG 1964 XXIII 525); its presence is therefore likely plesiomorphic in Pachycormidae.

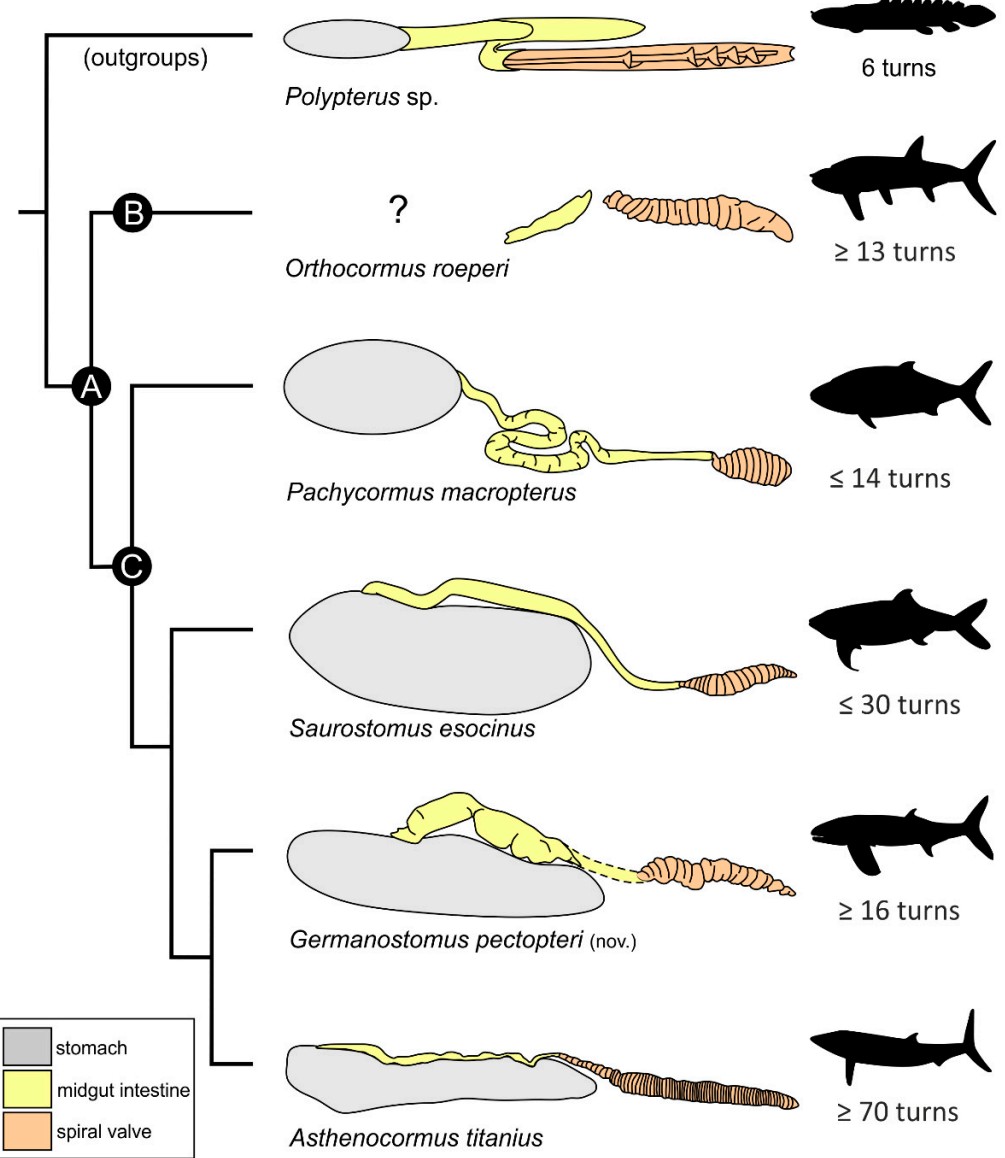

**Figure 10.** Phylogenetic framework of the gastrointestinal anatomy (GIT) in Pachycormiformes, excluding taxa in which the spiral valve is not preserved. *Polypterus* sp.: GIT reconstruction and silhouette redrawn and modified from Reference [88]. *Pachycormus macropterus*, *Saurostomus esocinus*, and *Germanostomus pectopteri* gen. et sp. nov.: GIT reconstructions based on first-hand observations of specimens. *Orthocormus roeperi*: GIT reconstruction drawn from fig.6 of Reference [14]. *Asthenocormus titanius* from Reference [76], fig 8 with details supplemented from Reference [109]. Gastrointestinal tracts not drawn to scale.

The size and placement of the spiral valve is highly variable between different osteichthyian clades. It is placed anterior to the intestine in most chondrichthyans, but it is generally situated closer to or immediately before the anus in sarcopterygians and actinopterygians [88,115,116,118–121]. The spiral valve in *Saurostomus esocinus* is elongated and placed high up in the abdominal cavity at some distance from the anus or ventral margin ([24]: figure 2). In *Pachycormus*, the spiral valve is much shorter and deeper, being set closer to the ventral margin and terminating closer to the anus (assumed to lie between the preanal scutes and the anal fin). The spiral valve is also well elongated in *Germanostomus*, although the position of the final rotation in relation to the anus is more similar to the condition in *Pachycormus* than to *Saurostomus*. The original position of the spiral valve relative to the ventral margin of the body cavity is unknown in *Germanostomus*. The number

of rotations is highly variable within pachycormids, with a maximum of 14 in *Pachycormus* (Figure 10), up to 30 in *S. esocinus* ([44]: figure 64; [106]: figure 4), and at minimum 16 in *Germanostomus pectopteri* gen. et sp. nov. According to Reference [109], the spiral valve of *Asthenocormus titanius* shows ≥70 rotations, likely reflecting the fish's increased body length compared to smaller pachycormids [1,88]. Individual rotations of the organ are deep and narrow in *Pachycormus* and *Asthencormus*, but wide and shallow in *Saurostomus* and *Germostomus* gen. nov. (Figure 10).

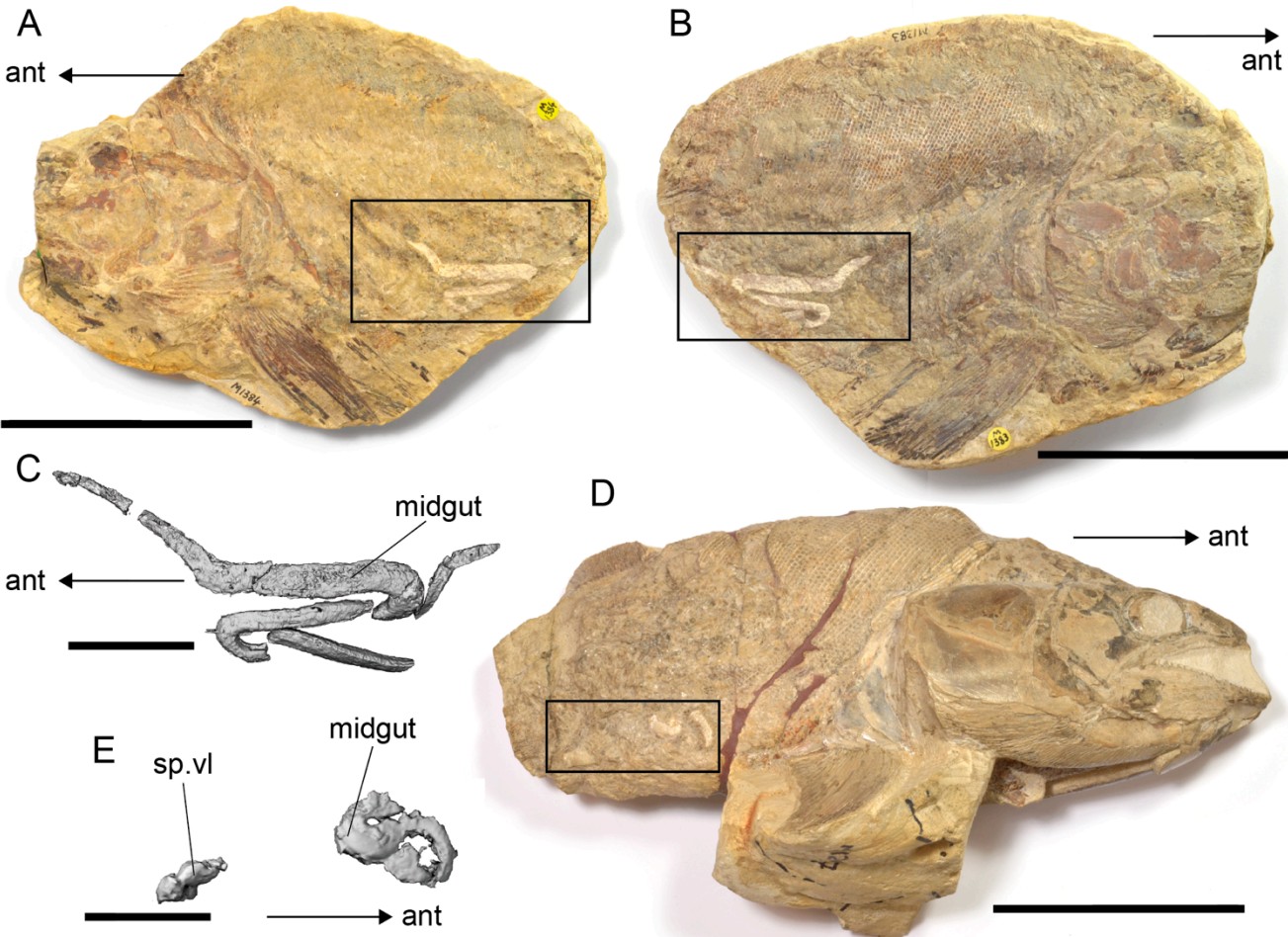

**Figure 11.** Alimentary canal in *Pachycormus* spp.: (**A**) Exposed midgut of *Pachycormus* in part (BRLSI.1384) and (**B**) counterpart (BRLSI.1383). (**C**) Three-dimensional render of the midgut with preserved regions of the alimentary tract in both part and counterpart combined. Note that some portions are entirely buried within the matrix and are not visible on the surface of the specimen. (**D**) Exposed midgut of *Pachycormus macropterus* (BRLSI.1297). (**E**) Three-dimensional render of the midgut and anterior portion of the spiral intestine. This latter feature is not visible on the surface of the specimen. The rectangular boxes in (**A**,**B**,**D**) indicate the extent of the preserved alimentary canal shown in the corresponding renders. sp.vl, Spiral valve. Scale bars = 50 (**A**,**B**,**D**); 25 mm (**C**,**E**). Photographs courtesy of Matt Williams (BRLSI).

The spiral valve in the extant *Polypterus* (Cladistia) is greatly elongated relative to the midgut, although no more than six turns are present, with each rotation of the mucosa being well spaced [108,124]. By contrast, the rotations in pachycormids, paddlefishes (6 turns), sturgeons (≤8 turns), †*Saurichthys* (30 turns [88]), *Amia* (4–5 turns), and lepisosteiforms (3–4 turns) are all tightly arranged and narrow through the entire length of the spiral valve [88,125,126]. The length of the spiral valve, or number of rotations, relative to body size is not generally correlated with diet [86] but rather reflects the (a) phylogenetic history

and (b) body size [88]. The higher number of turns in the spiral valve of *Germanostomus* when compared with *Pachycormus*, but falling far short of the count in *Asthenocormus*, is consistent with this pattern in Pachycormiformes, but given both the similar body size (1–2 m in total length) and close phylogenetic relatedness between *Pachycormus*, *Saurostomus*, and *Germanostomus*, additional factors must be considered.

The relationship between the number of turns in the spiral valve and midgut length has not previously been investigated. It is possible that the increased surface area of the folded midgut of *Pachycormus macropterus* may compensate for its unusually short spiral valve with fewer rotations, and that the rates of epithelial absorption were greater in the midgut region than in the hindgut. *Pachycormus*, to the best of our knowledge, is the only known stem-teleost to possess both a strongly folded midgut intestine and a spiral valve organ, although a similar morphology is known also for *Acipensor*. The midgut appears to be very short relative to the spiral valve in *Germanostomus*, although this may be a consequence of the ventrolateral orientation of the skeleton. Given the morphology in *Asthenocormus* and *Saurostomus*, it is reasonable to assume that the midgut attaches to the anterodorsal surface of the stomach in *Germanostomus* and, therefore, the exposed midgut length in SMNS 15815 might be severely underestimated.

## 5. Conclusions

We described a new genus and species of pachycormiform fish, *Germanostomus pectopteri* gen. et sp. nov., based on a near-complete, exceptionally well-preserved specimen from the Early Jurassic (Toarcian) Posidonienschiefer Formation of Holzmaden in SW Germany. Pachycormiformes is shown to be more diverse during its early radiation than previously considered. Phylogenetic analysis places *Germanostomus* gen. nov. between *Saurostomus esocinus* and *Ohmdenia multidentata*, near the base of Asthenocorminae and outside of the suspension-feeding clade. *G. pectopteri* shares those synapomorphies of the suspension-feeding clade which are already present in *Saurostomus*, namely, the absence of the infraorbitals and suborbitals, vertebral ossification restricted to the neural and haemal arches, and elongation of the jaws relative to skull length. Several additional synapomorphies of the suspension-feeding clade that are not present in *Saurostomus* are found in *Germanostomus pectopteri*, notably: (1) the absence of the opercular process on the hyomandibula; (2) increased elongation of the premaxilla and dentary; (3) further reduction of scales; (4) a pectoral fin that is wide and distally rounded (shared in *Ohmdenia* and *Leedsichthys*); (5) reduced ornamentation on the external skull bones. The macroevolutionary elongation of the jaws relative to the rest of the skull and the loss of the opercular process on the hyomandibula occurred early in asthenocormine evolution, with the acquisition of these characteristics dating from the *serpentinum* Zone of the early Toarcian. *Germanostomus* thus provides something of a crucial 'missing link' in the early evolution of suspension-feeding within Pachycormiformes. The gastrointestinal anatomy of *Pachycormus*, *Saurostomus*, and *Germanostomus* is described in detail and compared for the first time. All asthenocormines to the exclusion of *Pachycormus* appear to share a midgut beginning anterior and dorsal to the stomach; this could represent a synapomorphy in digestive anatomy uniting the clade. *Pachycormus* is the only known pachycormid to possess a folded midgut intestine, similar to the condition shared with *Acipenser* and superficially similar to derived teleosts. Gastrointestinal anatomy evidently holds both taxonomic and palaeoecological value in pachycormid research.

**Supplementary Materials:** The following supporting information can be downloaded at: https://www.mdpi.com/article/10.3390/d14121026/s1. Supplementary File S1 contains the following contents: (1) Supplementary Figures S1–S10, (2) Systematic description of the caudal fin of *Martillichthys renwickae* Liston, and (3) phylogenetic analysis including character lists and scoring: Supplementary Figures in Supplementary File S1: Figure S1 = Historic photograph of SMNS 15815 taken prior to 1944. Figure S2 = Caudal fin of Martillichthys renwickae Liston, NHMUK PV P. 61563. Figure S3 = Strict consensus of 12 trees based on the Friedman (2012) matrix. Figure S4 = Strict consensus tree from Friedman (2012) with Jackknife (36) applied. Figure S5 = Agreement subtree of the results from the

Friedman (2012) matrix. Figure S6 = Strict consensus of 5 trees based on the Gouiric–Cavalli and Arratia (2022) matrix. Figure S7 = Strict consensus tree based on the Gouiric–Cavalli and Arratia (2022) ma-trix. Figure S8 = Agreement subtree of consensus tree produced using the Gouiric–Cavalli and Arratia (2022) matrix. Figure S9 = Ohmdenia multidentata Holotype–GPIT-PV-31531. Figure S10 = Gastrointestinal anatomy in two examples of Pachycormus. Supplementary File S2 Updated TNT file of the Friedman (2012) phylogenetic analysis matrix, including scoring for *Germanostomus pectopteri* gen. et sp. nov. Supplementary File S3 TNT file of the Gouiric–Cavalli and Arratia (2022) phylogenetic analysis matrix, with updated scoring for *Saurostomus esocinus* and *Germanostomus pectopteri* gen. et sp. nov.

**Author Contributions:** Conceptualization, S.L.A.C. and E.E.M.; methodology, S.L.A.C. and E.E.M.; software, S.G.; formal analysis, S.L.A.C.; E.E.M.; S.G. and H.Y.; investigation, S.L.A.C.; E.E.M.; S.G. and H.Y.; resources, S.L.A.C.; E.E.M. and S.G.; data curation, S.L.A.C.; E.E.M. and S.G.; writing—original draft preparation, S.L.A.C.; E.E.M. and S.G.; writing—review and editing, S.L.A.C.; E.E.M. and S.G.; visualization, S.L.A.C. and S.G.; supervision, S.L.A.C.; E.E.M. and S.G.; project administration, S.L.A.C. All authors have read and agreed to the published version of the manuscript.

**Funding:** This research received no external funding.

**Institutional Review Board Statement:** Not applicable.

**Informed Consent Statement:** Not applicable.

**Data Availability Statement:** Not applicable.

**Acknowledgments:** Emma Bernard (NHMUK), Anne Krahl (GPIT), Rolf Hauff and Franziska Hauff (Urwelt Hauff), Adriana López-Arbarello (BSPG), and Matt Williams (BRLSI) are thanked for allowing us to examine comparative material in their collections. Matt Williams also provided high-quality photographs of the BRLSI specimens. Recognition must be given to †B. Hauff in retrospect for his excellent preparation of the holotype of *Germanosteus pectopteri* and to Lorie Barber for her skilled preparation of the *Pachycormus* Strawberry Bank material. We acknowledge the generosity of the original donors of the material (†R. Heilner and †T. Hermann) for whom without the new taxon would likely forever remain unknown to science. Thanks to Tom Davies, Benjamin Moon and Liz Martin-Silverstone for assistance with CT scanning. We also thank the two anonymous reviewers for their helpful comments. Thanks also to Jürgen Kriwet for his invitation to contribute to the Special Issue on 'Evolution and Diversity of Fishes in Deep Time'.

**Conflicts of Interest:** The authors declare no conflict of interest.

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
