# Peer review of "A New Large †Pachycormiform (Teleosteomorpha: †Pachycormiformes) from the Lower Jurassic of Germany, with Affinities to the Suspension-Feeding Clade, and Comments on the Gastrointestinal Anatomy of Pachycormid Fishes"

_diversity, doi:10.3390/d14121026_

Round 1
Reviewer 1 Report
Dear Authors,
this is a detailed and interesting work, which should be published after minor revisions.Attached a pdf with some comments, corrections and additions. Perhaps the rather long manuscript can be shortened a bit by restricting the comparisons to the discussion section and removing them from the description section of the new taxon. Are you sure you want to write Teleostei in the title of the manuscript? If so, I would explain that briefly in the text. According to Arratia 1999, 2013, 2015... the Pachycormiformes belong to the teleosteomorpha and not to the teleostei in the strict sense. For References see "Instructions for Authors" and adjust accordingly.

Author Response
Dear reviewer,
Thank you very much for taking the time to review our manuscript. Your comments were indeed very helpful.
We have addressed all of the comments made on the manuscript and corrected all the typos which you helpfully highlighted.
Following your comments, we have changed 'Teleostei' to 'Teleosteomorpha' in reference to Pachycormiformes. We have also made appropriate changes to the reference list to better fit the journal's style, as outlined in the 'Instructions for authors'.
You may be of interest to know just for future reference, the Munich specimen of Pseudoasthenocormus retrodorsalis (BSPG1956 I 361) is not the holotype. Eastman's type specimen, according to Lambers (1992) is CM.4863, housed in the Carnegie Museum, Pittsburgh. This is a common misconception which is often repeated in the literature.
Best wishes,
Reviewer 2 Report
Cooper and colleagues describe a new actinopterygian fish from the Lower Jurassic Poseidonienschiefer Formation of southern Germany.
It is a superb work reporting detailed systematic descriptions of spectacular specimens also having exceptionally preserved the intestinal tracts.
The results of this study represent a significant contribution to the knowledge of the Pachycormiformes, a Mesozoic group of large ray-finned fish that could have a crucial role in the holostean-teleostean transition.
The manuscript appears well organized and well written, and I have only a few suggestions and corrections (mainly typos) reported in an annotated pdf file.
I particular, recommend using the binomial nomenclature for species names used for biozones and subzones in accordance with the International Stratigraphic Guide (https://www.idigbio.org/wiki/images/7/7f/255-271_Murphy_.pdf ) which recommends compliance with the International Code of Zoological Nomenclature (Article 5.1: Principle of Binomial Nomenclature).
In conclusion, I strongly recommend the publication of this manuscript after a few minor corrections.

Author Response
Dear reviewer,
Thank you very much for taking the time to review our manuscript. Your comments were indeed very helpful.
We have addressed all of the comments made on the manuscript and corrected all the typos which you helpfully highlighted.
We have opted not to add generic names to the ammonite suzbzones because we have been informed by a colleague (Guenter Schweigert) that such a change would be unnecessary. Species name only is the more widely used scheme in previous published works on the Posidonienschiefer. We have asked the Editor for his decision on this, and will be happy to include them if the Editor asks for them.
Best wishes,